# BEER: Fast $O(1/T)$ Rate for Decentralized Nonconvex Optimization with Communication Compression

**Haoyu Zhao**
Princeton University
haoyu@princeton.edu

**Boyue Li**
Carnegie Mellon University
boyuel@andrew.cmu.edu

**Zhize Li**[*]
Carnegie Mellon University
zhizel@andrew.cmu.edu

**Peter Richtárik**
King Abdullah University of Science and Technology
peter.richtarik@kaust.edu.sa

**Yuejie Chi**
Carnegie Mellon University
yuejiec@andrew.cmu.edu

## Abstract

Communication efficiency has been widely recognized as the bottleneck for large-scale decentralized machine learning applications in multi-agent or federated environments. To tackle the communication bottleneck, there have been many efforts to design communication-compressed algorithms for decentralized nonconvex optimization, where the clients are only allowed to communicate a small amount of quantized information (aka bits) with their neighbors over a predefined graph topology. Despite significant efforts, the state-of-the-art algorithm in the nonconvex setting still suffers from a slower rate of convergence $O((G/T)^{2/3})$ compared with their uncompressed counterpart, where $G$ measures the data heterogeneity across different clients, and $T$ is the number of communication rounds. This paper proposes BEER, which adopts communication compression with gradient tracking, and shows it converges at a *faster rate* of $O(1/T)$. This significantly improves over the state-of-the-art rate, by matching the rate without compression even under arbitrary data heterogeneity. Numerical experiments are also provided to corroborate our theory and confirm the practical superiority of BEER in the data heterogeneous regime.

## 1   Introduction

Decentralized machine learning is gaining attention in both academia and industry because of its emerging applications in multi-agent systems such as the internet-of-things (IoT) and networked autonomous systems [31, 43]. One of the key problems in decentralized machine learning is on-device training, which aims to optimize a machine learning model using the datasets stored on (geographically) different clients, and can be formulated as a *decentralized optimization* problem.

Decentralized optimization aims to solve the following optimization problem without sharing the local datasets with other clients:

$$\min_{\boldsymbol{x} \in \mathbb{R}^d} \left\{ f(\boldsymbol{x}; \mathcal{D}) := \frac{1}{n} \sum_{i=1}^{n} f(\boldsymbol{x}; \mathcal{D}_i) \right\}, \tag{1}$$

where $f(\boldsymbol{x}; \mathcal{D}_i) := \mathbb{E}_{\xi_i \sim \mathcal{D}_i} f(\boldsymbol{x}; \xi_i)$ for $i \in [n]$, and $n$ is the total number of clients. Here, $\boldsymbol{x} \in \mathbb{R}^d$ is the machine learning model, $f(\boldsymbol{x}; \mathcal{D})$, $f(\boldsymbol{x}; \mathcal{D}_i)$, and $f(\boldsymbol{x}; \xi_i)$ denote the loss functions of the model $\boldsymbol{x}$ on the entire dataset $\mathcal{D}$, the local dataset $\mathcal{D}_i$, and a random data sample $\xi_i$, respectively. Different

---

[*]Corresponding author.

36th Conference on Neural Information Processing Systems (NeurIPS 2022).

from the widely studied distributed or federated learning setting where there is a central server to coordinate the parameter sharing across all clients, in the decentralized setting, each client can only communication with its neighbors over a communication network determined by a predefined network topology.

The main bottleneck of decentralized optimization—when it comes to large-scale machine learning applications—is communication efficiency, due to the large number of clients involved in the network [43] and the enormous size of machine learning models [4], exacerbated by resource constraints such as limited bandwidth availability and stringent delay requirements. One way to reduce the communication cost is communication compression, which only transmits compressed messages (with fewer bits) between the clients using *compression operators*. The compression operators come with many design choices and offer great flexibility in different trade-offs of communication and computation in practice. Even though communication compression has been extensively applied to distributed or federated optimization with a central server [47, 12, 6, 26, 11, 25, 40, 41, 27], its use in the decentralized setting has been relatively sparse. Most of the existing approaches only apply to the strongly convex setting [39, 17, 30, 19, 29, 23], and only a few consider the general nonconvex setting [16, 51, 45].

**Our contributions** This paper considers decentralized optimization with communication compression, focusing on the *nonconvex* setting due to its critical importance in modern machine learning, such as training deep neural networks [21], word embeddings, and other unsupervised learning models [42]. Unfortunately, existing algorithms [16, 51, 45] suffer from several important drawbacks in the nonconvex setting: they need strong bounded gradient or bounded dissimilarity assumptions to guarantee convergence, and the convergence rate is order-wise slower than their uncompressed counterpart in terms of the communication rounds (see Table 1).

In this paper, we introduce BEER, which is a decentralized optimization algorithm with communication compression using gradient tracking. BEER not only removes the strong assumptions required in all prior works, but enjoys a faster convergence rate in the nonconvex setting. Concretely, we have the following main contributions (see Tables 1 and 2).

1. We show that BEER converges at a fast rate of $O(1/T)$ in the nonconvex setting, which improves over the state-of-the-art rate $O(1/T^{2/3})$ of CHOCO-SGD [16] and Deepsqueeze [51], where $T$ is the number of communication rounds. This matches the rate without compression even under arbitrary data heterogeneity across the clients.

2. We also provide the analysis of BEER under the Polyak- Łojasiewicz (PL) condition (Assumption 4), and show that BEER converges at a linear rate (see Table 2). Note that strong convexity implies the PL condition, and thus BEER also achieves linear convergence in the strongly convex setting.

3. We run numerical experiments on real-world datasets and show BEER achieves superior or competitive performance when the data are heterogeneous compared with state-of-the-art baselines with and without communication compression.

To the best of our knowledge, BEER is the *first* algorithm that achieves $O(1/T)$ rate without the bounded gradient or bounded dissimilarity assumptions, supported by a strong empirical performance in the data heterogeneous setting.

**Notation** Throughout this paper, we use boldface letters to denote vectors, e.g., $\boldsymbol{x} \in \mathbb{R}^d$. Let $[n]$ denote the set $\{1, 2, \cdots, n\}$, $\mathbf{1}$ be the all-one vector, $\boldsymbol{I}$ be the identity matrix, $\|\cdot\|$ denote the Euclidean norm of a vector, and $\|\cdot\|_{\mathrm{F}}$ denote the Frobenius norm of a matrix. Let $\langle \boldsymbol{u}, \boldsymbol{v} \rangle$ denote the standard Euclidean inner product of two vectors $\boldsymbol{u}$ and $\boldsymbol{v}$. In addition, we use the standard order notation $O(\cdot)$ to hide absolute constants.

Due to space constraints, additional discussions of related work, further experiments, and the proof details can be found in the appendix.

## 2 Problem Setup

In this section, we formally define the decentralized optimization problem with communication compression, and introduce a few important quantities and assumptions that will be used in developing our algorithm and theory.

| Algorithm | Convergence rate | Strong assumption |
|---|---|---|
| SQuARM-SGD [45] | $O\left(\frac{1}{\sqrt{nT}} + \frac{nG^2}{T}\right)$ | Bounded Gradient |
| DeepSqueeze [51] | $O\left(\left(\frac{G}{T}\right)^{2/3}\right)$ | Bounded Dissimilarity |
| CHOCO-SGD [16] | $O\left(\left(\frac{G}{T}\right)^{2/3}\right)$ | Bounded Gradient |
| BEER (Algorithm 1) | $O\left(\frac{1}{T}\right)$ | — |

Table 1: Comparison of convergence rates for existing decentralized methods with communication compression in the nonconvex setting. Here, the parameter $G$ refers the quantity either in the bounded gradient assumption $\mathbb{E}_{\xi_i \sim \mathcal{D}_i} \|\nabla f(\boldsymbol{x}, \xi_i)\|^2 \leq G^2$ or the bounded dissimilarity assumption $\mathbb{E}_i \|\nabla f(\boldsymbol{x}, \mathcal{D}_i) - \nabla f(\boldsymbol{x}, \mathcal{D})\|^2 \leq G^2$, both of which are very strong assumptions (the bounded dissimilarity assumption is slightly weaker) that BEER does *not* require. All algorithms support the use of stochastic gradients with bounded local variance at local clients.

| Assumptions | Convergence rate | Theorem |
|---|---|---|
| $f_i$ is $L$-smooth | $\frac{1}{T}\sum_{t=0}^{T-1} \mathbb{E}\|\nabla f(\bar{\boldsymbol{x}}^t)\|^2 \leq \frac{2(\Phi_0 - \Phi_T)}{\eta T}$ | Theorem 1 |
| $f_i$ is $L$-smooth
$f$ satisfies PL condition | $\Phi_T \leq (1 - \mu\eta)^T \Phi_0$ | Theorem 3 |

Table 2: Summary of the established convergence rates for the proposed BEER algorithm in the nonconvex setting. Here, $\bar{\boldsymbol{x}}^t$ is the average model of all clients, $\eta$ is the step size, $\Phi_t$ is the Lyapunov function (cf. (4)), and $\mu$ is the PL-condition parameter (cf. Assumption 4). We do not assume the bounded gradient or bounded dissimilarity assumption.

**Decentralized optimization** The goal of decentralized optimization is to solve

$$\min_{\boldsymbol{x} \in \mathbb{R}^d} \left\{ f(\boldsymbol{x}) := \frac{1}{n}\sum_{i=1}^n f_i(\boldsymbol{x}) \right\},$$

where $n$ is the number of clients, $f(\boldsymbol{x})$ is the global objective function, and $f_i(\boldsymbol{x}) := f(\boldsymbol{x}; \mathcal{D}_i) := \mathbb{E}_{\xi_i \sim \mathcal{D}_i} f(\boldsymbol{x}; \xi_i)$ is the local objective function, with $\boldsymbol{x}$ the parameter of interest, $\xi_i$ a random data sample drawn from the local dataset $\mathcal{D}_i$.

In the decentralized setting, the clients can only communicate with their local neighbors over a prescribed network topology, which is specified by an undirected weighted graph $\mathcal{G}([n], E)$. Here, each node in $[n]$ represents a client, and $E$ is the set of possible communication links between different clients. Information sharing across the clients is implemented mathematically by the use of a mixing matrix $\boldsymbol{W} = [w_{ij}] \in [0,1]^{n \times n}$, which is defined in accordance with the network topology: we assign a positive weight $w_{ij}$ for any $(i,j) \in E$ and $w_{ij} = 0$ for all $(i,j) \notin E$. We make the following standard assumption on the mixing matrix [36].

**Assumption 1 (Mixing matrix)** *The mixing matrix $\boldsymbol{W} = [w_{ij}] \in [0,1]^{n \times n}$ is symmetric ($\boldsymbol{W}^\top = \boldsymbol{W}$) and doubly stochastic ($\boldsymbol{W}\mathbf{1} = \mathbf{1}, \mathbf{1}^\top \boldsymbol{W} = \mathbf{1}^\top$). Let its eigenvalues be $1 = |\lambda_1(\boldsymbol{W})| > |\lambda_2(\boldsymbol{W})| \geq \cdots \geq |\lambda_n(\boldsymbol{W})|$. The spectral gap is denoted by*

$$\rho := 1 - |\lambda_2(\boldsymbol{W})| \in (0, 1]. \tag{2}$$

The spectral gap of a mixing matrix is closely related to the network topology, see Nedić et al. [36] for its scaling with respect to the network size (i.e. the number of clients $n$) for representative network topologies.

**Compression operators** Compression, in the forms of quantization or sparsification, can be used to reduce the total communication cost. We now introduce the notion of a randomized *compression operator*, which is widely used in the decentralized/federated optimization literature, e.g. Tang et al. [49], Stich et al. [47], Koloskova et al. [16], Richtárik et al. [40], Fatkhullin et al. [8].

**Definition 1 (Compression operator)** *A randomized map* $\mathcal{C} : \mathbb{R}^d \mapsto \mathbb{R}^d$ *is an* $\alpha$-*compression operator if for all* $\boldsymbol{x} \in \mathbb{R}^d$, *it satisfies*

$$\mathbb{E}\left[\|\mathcal{C}(\boldsymbol{x}) - \boldsymbol{x}\|^2\right] \leq (1 - \alpha)\|\boldsymbol{x}\|^2. \tag{3}$$

*In particular, no compression* ($\mathcal{C}(\boldsymbol{x}) \equiv \boldsymbol{x}$) *implies* $\alpha = 1$.

Compared with the *unbiased compression operator* used in, e.g., Alistarh et al. [1], Khirirat et al. [15], Mishchenko et al. [33], Li and Richtárik [24], the compression operator in Definition 1 does not impose the additional constraint on the expectation such that $\mathbb{E}[\mathcal{C}(\boldsymbol{x})] = \boldsymbol{x}$. Besides, it is always possible to convert an *unbiased compression operator* into a biased one satisfying Definition 1, and thus Definition 1 is a generalization of the unbiased compression operator that allows *biased* compression. Practical examples of the compression operators are provided in Appendix B.

**Assumptions on functions** We now state the assumptions on the functions $\{f_i\}$ and $f$. Throughout this paper, we assume that $f^* = \min_{\boldsymbol{x}} f(\boldsymbol{x})$ exists and $f^* > -\infty$.

In the nonconvex setting, we assume that the functions $\{f_i\}_{i \in [n]}$ are arbitrary functions that satisfy the following standard smoothness assumption.

**Assumption 2 (Smoothness)** *The function* $f$ *is* $L$-*smooth if there exists* $L \geq 0$ *such that*

$$\|\nabla f(\boldsymbol{x}_1) - \nabla f(\boldsymbol{x}_2)\| \leq L \|\boldsymbol{x}_1 - \boldsymbol{x}_2\|, \forall \boldsymbol{x}_1, \boldsymbol{x}_2 \in \mathbb{R}^d.$$

In addition, we allow local computation to be performed via stochastic gradient updates, where $\tilde{\nabla} f_i(\boldsymbol{x}) := \nabla f_i(\boldsymbol{x}; \xi_i)$ denotes a local stochastic gradient computed via a sample $\xi_i$ drawn i.i.d. from $\mathcal{D}_i$, and $\tilde{\nabla}_b f_i(\boldsymbol{x}) := \frac{1}{b} \sum_{j=1}^{b} \nabla f_i(\boldsymbol{x}; \xi_{i,j})$ denotes the stochastic gradient computed by a minibatch with size $b$ drawn i.i.d. from $\mathcal{D}_i$. We assume $\tilde{\nabla} f_i(\boldsymbol{x})$ and $\tilde{\nabla}_b f_i(\boldsymbol{x})$ have bounded variance, which is again standard in the decentralized/federated optimization literature [32, 13, 16].

**Assumption 3 (Bounded variance)** *There exists a constant* $\sigma \geq 0$ *such that for all* $i \in [n]$ *and* $\boldsymbol{x} \in \mathbb{R}^d$,

$$\mathbb{E}\left\|\tilde{\nabla} f_i(\boldsymbol{x}) - \nabla f_i(\boldsymbol{x})\right\|^2 \leq \sigma^2.$$

*For a stochastic gradient with minibatch size* $b$, *we have*

$$\mathbb{E}\left\|\tilde{\nabla}_b f_i(\boldsymbol{x}) - \nabla f_i(\boldsymbol{x})\right\|^2 \leq \frac{\sigma^2}{b}.$$

In addition, we consider the setting when the function $f$ additionally satisfies the following Polyak-Łojasiewicz (PL) condition [37], which can lead to fast linear convergence even when the function is nonconvex.

**Assumption 4 (PL condition)** *There exists some constant* $\mu > 0$ *such that for any* $\boldsymbol{x} \in \mathbb{R}^d$,

$$\|\nabla f(\boldsymbol{x})\|^2 \geq 2\mu(f(\boldsymbol{x}) - f^*).$$

Note that the PL condition is a weaker assumption than strong convexity, which means that if the objective function $f$ is $\mu$-strongly convex, then the PL condition also holds with the parameter $\mu$.

## 3 Proposed Algorithm

In this section, we introduce our proposed algorithm BEER for decentralized nonconvex optimization with compressed communication. Before embarking on the description of BEER, we introduce some convenient matrix notation. Since in a decentralized setting, the parameter estimates at different clients are typically different, we use $\boldsymbol{X} = [\boldsymbol{x}_1, \boldsymbol{x}_2, \ldots, \boldsymbol{x}_n]$ to denote the collection of parameter estimates from all clients, where $\boldsymbol{x}_i$ is from client $i$. The average of $\{\boldsymbol{x}_i\}_{i \in [n]}$ is denoted by $\bar{\boldsymbol{x}} := \frac{1}{n} \boldsymbol{X} \mathbf{1}$. Other quantities are defined similarly. With slight abuse of notation, we define

$$\nabla F(\boldsymbol{X}) := [\nabla f_1(\boldsymbol{x}_1), \nabla f_2(\boldsymbol{x}_2), \ldots, \nabla f_n(\boldsymbol{x}_n)] \in \mathbb{R}^{d \times n},$$

---

**Algorithm 1** BEER: BEtter comprEssion for decentRalized optimization

---

1: **Input:** Initial point $X^0 = x_0 1^\top$, $G^0 = 0$, $H^0 = 0$, $V^0 = \nabla F(X_0)$, step size $\eta$, mixing step size $\gamma$, minibatch size $b$.
2: **for** $t = 0, 1, \ldots$ **do**
3:     $X^{t+1} = X^t + \gamma H^t(W - I) - \eta V^t$
4:     $Q_h^{t+1} = \mathcal{C}(X^{t+1} - H^t)$      // client $i$ sends $q_{h,i}^{t+1}$ to all its neighbors
5:     $H^{t+1} = H^t + Q_h^{t+1}$
6:     $V^{t+1} = V^t + \gamma G^t(W - I) + \tilde{\nabla}_b F(X^{t+1}) - \tilde{\nabla}_b F(X^t)$
7:     $Q_g^{t+1} = \mathcal{C}(V^{t+1} - G^t)$      // client $i$ sends $q_{g,i}^{t+1}$ to all its neighbors
8:     $G^{t+1} = G^t + Q_g^{t+1}$
9: **end for**

---

which collects the local gradients computed at the local parameters. Similarly, the stochastic variant is defined as $\tilde{\nabla}_b F(X) := [\tilde{\nabla}_b f_1(x_1), \tilde{\nabla}_b f_2(x_2), \ldots, \tilde{\nabla}_b f_n(x_n)]$. We also allow the compression operator to take vector values, which are applied in a column-wise fashion, i.e., $\mathcal{C}(X) := [\mathcal{C}(x_1), \ldots, \mathcal{C}(x_n)] \in \mathbb{R}^{d \times n}$.

We now proceed to describe BEER, which is detailed in Algorithm 1 using the matrix notation introduced above. At the $t$-th iteration, BEER maintains the current model estimates $X^t$ and the global gradient estimates $V^t$ across the clients. At the crux of its design, BEER also tracks and maintains two control sequences $H^t$ and $G^t$ that serve as compressed surrogates of $X^t$ and $V^t$, respectively. In particular, these two control sequences are updated by aggregating the received compressed messages alone (cf. Line 5 and Line 8).

It then boils down to how to carefully update these quantities in each iteration with communication compression, which we now explain in details. To begin, note that for each client $i$, BEER not only maintains its own parameters $\{x_i^t, v_i^t, h_i^t, g_i^t\}$, but also the control variables from its neighbors, namely, $\{h_j^t\}_{j \in \mathcal{N}(i)}$ and $\{g_j^t\}_{j \in \mathcal{N}(i)}$.

**Update the model estimate** Each client $i$ first updates its model $x_i^{t+1}$ according to Line 3. By thinking of $\{h_j^t\}_{j \in \mathcal{N}(i)}$ as a surrogate of $\{x_j^t\}_{j \in \mathcal{N}(i)}$, the second term aims to achieve better consensus among the clients through mixing, while the last term performs a gradient descent update.

**Update the global gradient estimate** Each client $i$ updates the global gradient estimate $v_i^{t+1}$ according to Line 6, where the last correction term—based on the difference of the gradients at consecutive models—is known as a trick called *gradient tracking* [38, 7, 35]. The use of gradient tracking is critical: as shall be seen momentarily, it contributes to the key difference from CHOCO-SGD that enables the fast rate of $O(1/T)$ without any bounded dissimilarity or bounded gradient assumptions. Indeed, if we remove the control sequence $G^t$ and substitute Lines 6-8 by $V^{t+1} = \tilde{\nabla}_b F(X^{t+1})$, we recover CHOCO-SGD from BEER.

**Update the compressed surrogates with communication** To update $\{h_j^t\}_{j \in \mathcal{N}(i)}$, each client $i$ first computes a compressed message $q_{h,i}^{t+1}$ that encodes the difference $x_i^{t+1} - h_i^t$, and broadcasts to its neighbors (cf. Line 4). Then, each client $i$ updates $\{h_j^t\}_{j \in \mathcal{N}(i)}$ by aggregating the received compressed messages $\{q_{h,j}^{t+1}\}_{j \in \mathcal{N}(i)}$ following Line 5. The updates of $\{g_j^t\}_{j \in \mathcal{N}(i)}$ can be performed similarly. Moreover, all the compressed messages can be sent in a single communication round at one iteration, i.e., the communications in Lines 4 and 7 can be performed at once. This leverages EF21 [40] for communication compression, which is a *better and simpler* algorithm that deals with biased compression operators compared with the error feedback (or error compensation, EF/EC) framework [12, 46]. Using the control sequence $G^t$, BEER does not need to apply EF/EC explicitly and can deal with the error implicitly.

## 4 Convergence Guarantees

In this section, we show the convergence guarantees of BEER under different settings: the $O(1/T)$ rate in the nonconvex setting in Section 4.1, and the improved linear rate under the PL condition (Assumption 4) in Section 4.2. In Section 4.3, we briefly sketch the proof.

Our convergence guarantees are based on an appropriately designed Lyapunov function, given by

$$\Phi_t = \mathbb{E} f(\bar{\boldsymbol{x}}^t) - f^* + \frac{c_1 L}{n}\Omega_1^t + \frac{c_2\rho^2}{nL}\Omega_2^t + \frac{c_3 L}{n}\Omega_3^t + \frac{c_4\rho^4}{nL}\Omega_4^t, \tag{4}$$

where the choice of constants $\{c_i\}_{i=1}^4$ might be different from theorem to theorem, $\mathbb{E} f(\bar{\boldsymbol{x}}^t) - f^*$ represents the sub-optimality gap, and the errors $\{\Omega_i^t\}_{i=1}^4$ are defined by

$$\Omega_1^t := \mathbb{E} \left\| \boldsymbol{H}^t - \boldsymbol{X}^t \right\|_{\mathrm{F}}^2, \quad \Omega_2^t := \mathbb{E} \left\| \boldsymbol{G}^t - \boldsymbol{V}^t \right\|_{\mathrm{F}}^2, \tag{5}$$

$$\Omega_3^t := \mathbb{E} \left\| \boldsymbol{X}^t - \bar{\boldsymbol{x}}^t \mathbf{1}^\top \right\|_{\mathrm{F}}^2, \quad \Omega_4^t := \mathbb{E} \left\| \boldsymbol{V}^t - \bar{\boldsymbol{v}}^t \mathbf{1}^\top \right\|_{\mathrm{F}}^2.$$

Here, $\Omega_1^t$ and $\Omega_2^t$ denote the compression errors for $\boldsymbol{X}^t$ and $\boldsymbol{V}^t$ when approximated using the compressed surrogates $\boldsymbol{H}^t$ and $\boldsymbol{G}^t$ respectively, and $\Omega_3^t$ and $\Omega_4^t$ denote the consensus errors of $\boldsymbol{X}^t$ and $\boldsymbol{V}^t$.

## 4.1 Convergence in the nonconvex setting

First, we present the following convergence result of BEER in the nonconvex setting when there is no local variance ($\sigma^2 = 0$), i.e., we can use the local full gradient $\nabla F(\boldsymbol{X}^t)$ instead of $\tilde{\nabla}_b F(\boldsymbol{X}^t)$ in Line 6 of Algorithm 1.

**Theorem 1 (Convergence in the nonconvex setting without local variance)** *Suppose Assumptions 1, and 2 hold, and we can compute the local full gradient $\nabla f_i(\boldsymbol{x})$ for any $\boldsymbol{x}$. Then there exist absolute constants $c_1, c_2, c_3, c_4, c_\gamma, c_\eta > 0$, such that if we set $\gamma = c_\gamma \alpha \rho$, $\eta = c_\eta \gamma \rho^2 / L$, then for the Lyapunov function $\Phi_t$ in (4), it holds*

$$\frac{1}{T} \sum_{t=0}^{T-1} \mathbb{E} \left\| \nabla f(\bar{\boldsymbol{x}}^t) \right\|^2 \leq \frac{2(\Phi_0 - \Phi_T)}{\eta T}.$$

Theorem 1 shows that BEER converges at a rate of $O(1/T)$ when there is no local variance ($\sigma^2 = 0$), which is faster than the $O(1/T^{2/3})$ rate by CHOCO-SGD [16] and DeepSqueeze [51], and the $O(1/\sqrt{T})$ rate by SQuARM-SGD [45]; see also Table 1.

More specifically, to achieve $\frac{1}{T} \sum_{t=0}^{T-1} \mathbb{E} \left\| \nabla f(\bar{\boldsymbol{x}}^t) \right\|^2 \leq \epsilon^2$, BEER needs $O\left(\frac{1}{\rho^3 \alpha \epsilon^2}\right)$ iterations or communication rounds, where $\rho$ and $\alpha$ are the spectral gap (cf. (2)) and the compression parameter (cf. (3)), respectively. In comparison, the state-of-the-art algorithm CHOCO-SGD [16] converges at a rate of $O((G/\rho^2 \alpha T)^{2/3})$, which translates to an iteration complexity of $O\left(\frac{G}{\rho^2 \alpha \epsilon^3}\right)$, with $G$ being the bounded gradient parameter, namely, $\mathbb{E}_{\xi_i \sim \mathcal{D}_i} \left\| \nabla f(\boldsymbol{x}, \xi_i) \right\|^2 \leq G^2$. Therefore, BEER improves over CHOCO-SGD not only in terms of a better dependency on $\epsilon$, but also removing the bounded gradient assumption, which is significant since in practice, $G$ can be excessively large due to data heterogeneity across the clients.

The dependency on $\alpha$ of BEER is consistent with other compression schemes, such as CHOCO-SGD, DeepSqueeze and SQuARM-SGD for the nonconvex setting, as well as LEAD [30] and EF-C-GT [29] for the strongly convex setting.

As for the dependency on $\rho$, BEER is slightly worse than CHOCO-SGD, where CHOCO-SGD has a dependency of $O(1/\rho^2)$ whereas BEER has a dependency of $O(1/\rho^3)$. This degeneration is also seen in the analysis of uncompressed decentralized algorithms using gradient tracking [48, 54], where the rate $O(1/\rho^2)$ is worse than the rate of $O(1/\rho)$ for basic decentralized SGD algorithms [14, 28] by a factor of $\rho$. In addition, both BEER and CHOCO-SGD use small mixing step size $\gamma$ to guarantee convergence, which makes the dependency on $\rho$ worse than their uncompressed counterparts.

**Stochastic gradient oracles** BEER also supports the use of stochastic gradient oracles with bounded local variance (Assumption 3). More specifically, we have the following theorem, which generalizes Theorem 1.

**Theorem 2 (Convergence in the nonconvex setting)** *Suppose Assumptions 1, 2 and 3 hold. Then there exist absolute constants $c_1, c_2, c_3, c_4, c_\gamma, c_\eta > 0$, such that if we set $\gamma = c_\gamma \alpha \rho$, $\eta = c_\eta \gamma \rho^2 / L$,*

| Algorithm | Communication rounds | Gradient complexity |
|---|---|---|
| SQuARM-SGD [45] | $O\left(\frac{nG^2}{\epsilon^2} + \frac{\sigma^2}{bn\epsilon^4}\right)$ | $O\left(\frac{\sigma^2}{n\epsilon^4} + \frac{nG^2}{\epsilon^2}\right)$ |
| DeepSqueeze [51] | $O\left(\frac{G}{\epsilon^3} + \frac{\sigma^2}{bn\epsilon^4}\right)$ | $O\left(\frac{\sigma^2}{n\epsilon^4} + \frac{G}{\epsilon^3}\right)$ |
| CHOCO-SGD [16] | $O\left(\frac{G}{\epsilon^3} + \frac{\sigma^2}{bn\epsilon^4}\right)$ | $O\left(\frac{\sigma^2}{n\epsilon^4} + \frac{G}{\epsilon^3}\right)$ |
| BEER (Algorithm 1) | $O\left(\frac{1}{\epsilon^2}\right)$ | $O\left(\frac{\sigma^2}{\epsilon^4} + \frac{1}{\epsilon^2}\right)$ |

Table 3: A more detailed comparison of the communication complexity and the gradient complexity with existing decentralized stochastic methods in the nonconvex setting to reach $\epsilon$-accuracy. Here, $G$ again measures the bounded gradient or bounded dissimilarity assumption, $\sigma^2$ and $b$ denote the gradient variance and batch size respectively. We omit the dependency on the compression ratio and the network topology parameter for brevity.

*then for the Lyapunov function $\Phi_t$ in (4), it holds*

$$\frac{1}{T}\sum_{t=0}^{T-1}\mathbb{E}\left\|\nabla f(\bar{\boldsymbol{x}}^t)\right\|^2 \leq \frac{2(\Phi_0 - \Phi_T)}{\eta T} + \frac{6c_4\sigma^2}{c_\gamma b\alpha L}.$$

In the presence of local variance, the squared gradient norm of BEER has an additional term that scales on the order of $O\left(\frac{\sigma^2}{\alpha b}\right)$ (ignoring other parameters). By choosing a sufficiently large minibatch size $b$, i.e. $b \geq O\left(\frac{\sigma^2}{\alpha\epsilon^2}\right)$, BEER maintains the iteration complexity $O\left(\frac{1}{\rho^3\alpha\epsilon^2}\right)$ to reach $\frac{1}{T}\sum_{t=0}^{T-1}\mathbb{E}\left\|\nabla f(\bar{\boldsymbol{x}}^t)\right\|^2 \leq \epsilon^2$, without the bounded gradient assumption, thus inheriting similar advantages over CHOCO-SGD as discussed earlier.

While our focus is on communication efficiency, to gain more insights, Table 3 summarizes both the communication rounds and the gradient complexity for different decentralized stochastic methods. While BEER does not require the bounded gradient assumption, it may lead to a worse gradient complexity in the data homogeneous setting due to the use of large minibatch size. Fortunately, this only impacts the local computation cost, and does not exacerbate the communication complexity, which is often the bottleneck. It is of great interest to further refine the design and analysis of BEER in terms of the gradient complexity.

### 4.2 Linear convergence with PL condition

Now, we show that the convergence of BEER can be strengthened to a linear rate with the addition of the PL condition (Assumption 4). Similar to the nonconvex setting, we first show the convergence result without local gradient variance ($\sigma^2 = 0$).

**Theorem 3 (Convergence under the PL condition without local variance)** *Suppose Assumptions 1, 2, and 4 hold, and we can compute the local full gradient $\nabla f_i(\boldsymbol{x})$ for any $\boldsymbol{x}$. Then there exist constants $c_1, c_2, c_3, c_4, c_\gamma, c_\eta > 0$, such that if we set $\gamma = c_\gamma\alpha\rho$, $\eta = c_\eta\gamma\rho^2/L$, then for the Lyapunov function $\Phi_t$ in (4), it holds*

$$\Phi_T \leq (1 - \mu\eta)^T\Phi_0.$$

Theorem 3 demonstrates that under the PL condition, BEER converges linearly to the global optimum $f^*$, where it finds an $\epsilon$-optimal solution in $O\left(\frac{\kappa}{\rho^3\alpha}\log\left(\frac{1}{\epsilon}\right)\right)$ iterations, with $\kappa := L/\mu$ the condition number. Note that in the strongly convex case, Liao et al. [29] proposed an algorithm that also uses error feedback compression and gradient tracking simultaneously, which achieves a linear rate of convergence with unclear dependencies with salient problem parameters. In comparison, BEER achieves an explicit linear rate of convergence in the strongly convex case as well, given strong convexity implies the PL condition. In fact, our analysis for the nonconvex setting—as will be made evident in our proof—almost implies immediately the linear convergence under the PL condition, thus provides a major step forward compared with prior analyses that only considered the strongly convex setting.

**Stochastic gradient oracles** Under the PL condition, BEER also supports the use of stochastic gradient oracles with bounded local variance (Assumption 3). The following theorem shows that BEER linearly converges to a neighborhood of size $O\left(\frac{\sigma^2}{\alpha b}\right)$ around the global optimum.

**Theorem 4 (Convergence under PL condition)** *Suppose Assumptions 1, 2, 3, and 4 hold. Then there exist absolute constants $c_1, c_2, c_3, c_4, c_\gamma, c_\eta > 0$, such that if we set $\gamma = c_\gamma \alpha \rho$, $\eta = c_\eta \gamma \rho^2 / L$, then for the Lyapunov function $\Phi_t$ in (4), it holds*

$$\Phi_T \leq (1 - \mu\eta)^T \Phi_0 + \frac{6c_4\sigma^2}{c_\gamma Lb\alpha}.$$

### 4.3 Proof sketch

We now provide a proof sketch of Theorem 1, which establishes the $O(1/T)$ rate of BEER in the nonconvex setting using full gradient, highlighting the technical reason of the rate improvement of BEER over CHOCO-SGD. The full proofs of our theorems are delegated to Appendix D.

Recalling the quantities $\Omega_1^t$ to $\Omega_4^t$ from (5), which capture the approximation errors using compression and the consensus errors of $\boldsymbol{X}^t$ and $\boldsymbol{V}^t$, we would like to control these errors by obtaining inequalities of the form:

$$\Omega_i^{t+1} \leq (1 - a_i)\Omega_i^t + b_i, \quad \forall i \in \{1, 2, 3, 4\},$$

where $0 < a_i < 1$ denotes the size of the contraction, and $b_i > 0$ wraps together other terms which may be dependent on $\Omega_j^t$ for $j \neq i$ as well as the expected squared gradient norm of $\bar{\boldsymbol{v}}^t$, i.e.,

$$\Omega_5^t = \mathbb{E}\left\|\bar{\boldsymbol{v}}^t\right\|^2. \tag{6}$$

Then, by choosing the Lyapunov function properly (cf. (4)), we can show that the Lyapunov function actually descends, and small manipulations lead to the claimed convergence results in Theorem 1.

We now explain briefly how gradient tracking helps in BEER. Note that CHOCO-SGD also has the control variable $\boldsymbol{H}^t$ for the model $\boldsymbol{X}^t$, therefore in its analysis, it also deals with the quantities $\Omega_1^t$ and $\Omega_3^t$. However, CHOCO-SGD also needs to bound the term $\|\boldsymbol{V}^t\|_{\mathrm{F}}^2$, where $\boldsymbol{V}^t = \nabla F(\boldsymbol{X}^t)$ for CHOCO-SGD when using full gradients. Thus, CHOCO-SGD needs to assume the bounded gradient assumption and only obtain a slower $O(1/T^{2/3})$ convergence rate. In contrast, BEER deals with the term $\|\boldsymbol{V}^t\|_{\mathrm{F}}^2$ by decomposing it using Young's inequality, leading to

$$\left\|\boldsymbol{V}^t\right\|_{\mathrm{F}}^2 \leq (1 + \beta)(\Omega_4^t)^2 + (1 + 1/\beta)(\Omega_5^t)^2$$

for some $\beta > 0$. Here, $\Omega_4^t$ can be controlled via the *gradient tracking* technique (see Line 6 in Algorithm 1) *without* the bounded gradient assumption, and $\Omega_5^t$ can be handled using the smoothness assumption (Assumption 2).

## 5 Numerical Experiments

This section presents numerical experiments on real-world datasets to showcase BEER's superior ability to handle data heterogeneity across the clients, by running each experiment on unshuffled datasets and comparing the performances with the state-of-the-art baseline algorithms both with and without communication compression. The code can be accessed at:



`https://github.com/liboyue/beer`.



**Experiment setup** We run experiments on two nonconvex problems to compare with the baseline algorithms both with and without communication compression: logistic regression with a nonconvex regularizer [52] on the `a9a` dataset [5], and training a 1-hidden layer neural network on the `MNIST` dataset [20].

For logistic regression with a nonconvex regularizer, following Wang et al. [52], the objective function over a datum $(\boldsymbol{a}, b)$ is defined as

$$f(\boldsymbol{x}; (\boldsymbol{a}, b)) = \log\left(1 + \exp(-b\boldsymbol{a}^\top \boldsymbol{x})\right) + \alpha \sum_{j=1}^{d} \frac{x_j^2}{1 + x_j^2},$$

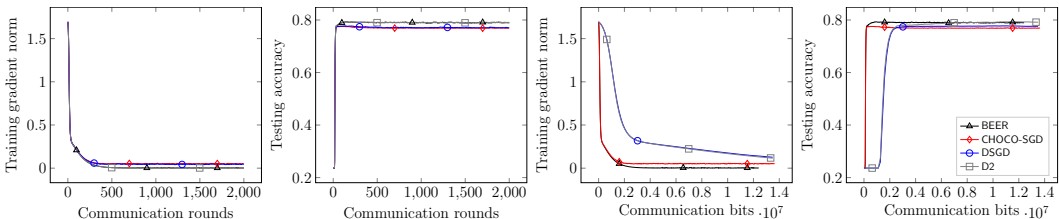

Figure 1: The training gradient norm and testing accuracy against communication rounds (left two panels) and communication bits (right two panels) for logistic regression with nonconvex regularization on unshuffled `a9a` dataset. Both BEER and CHOCO-SGD employ the biased $\text{gsgd}_b$ compression [1] with $b = 5$.

where the last term is the nonconvex regularizer and the regularization parameter is set to $\alpha = 0.05$.

For 1-hidden layer neural network training, we use 32 hidden neurons, sigmoid activation functions and cross-entropy loss. The objective function over a datum $(\boldsymbol{a}, b)$ is defined as

$$f(\boldsymbol{x}; (\boldsymbol{a}, b)) = \ell(\text{softmax}(\boldsymbol{W}_2 \, \text{sigmoid}(\boldsymbol{W}_1 \boldsymbol{a} + \boldsymbol{c}_1) + \boldsymbol{c}_2), b),$$

where $\ell(\cdot, \cdot)$ denotes the cross-entropy loss, the optimization variable is collectively denoted by $\boldsymbol{x} = \text{vec}(\boldsymbol{W}_1, \boldsymbol{c}_1, \boldsymbol{W}_2, \boldsymbol{c}_2)$, where the dimensions of the network parameters $\boldsymbol{W}_1, \boldsymbol{c}_1, \boldsymbol{W}_2, \boldsymbol{c}_2$ are $64 \times 784$, $64 \times 1$, $10 \times 64$, and $10 \times 1$, respectively.

For both experiments, we split the *unshuffled* datasets evenly to 10 clients that are connected by a ring topology. By using unshuffled data, we can simulate the scenario with high data heterogeneity across clients. Approximately, for the `a9a` dataset, 5 clients receive data with label 1 and others receive data with label 0; for the `MNIST` dataset, client $i$ receives data with label $i$. We use the FDLA matrix [53] as the mixing matrix to perform weighted information aggregation to accelerate convergence.

**Results**  We compare BEER with 1) CHOCO-SGD [17], which is the state-of-the-art nonconvex decentralized optimizing algorithm using communication compression, and 2) DSGD [28] and $D^2$ [50], which are decentralized optimization algorithms without compression. All algorithms are initialized in the same experiment by the same initial point. Moreover, we use the same best-tuned learning rate $\eta = 0.1$, batch size $b = 100$, and biased compression operator ($\text{gsgd}_b$) [1] for BEER and CHOCO-SGD on both experiments.

Figure 1 and Figure 2 plot the training gradient norm and testing accuracy against communication rounds and communication bits for logistic regression with nonconvex regularization and 1-hidden-layer neural network training, respectively.

In the nonconvex logistic regression experiment (cf. Figure 1), the algorithms with communication compression (BEER and CHOCO-SGD) converge faster than the uncompressed algorithms (DSGD and $D^2$) in terms of the communication bits. However, CHOCO-SGD fails to converge to a small gradient norm solution since it cannot tolerate a high level of data dissimilarity across different clients, and its performance is not comparable to $D^2$. In contrast, BEER can converge to a point with a relatively smaller gradient norm, which is comparable to $D^2$. The performance of testing accuracy is similar to that of the training gradient norm, where BEER achieves the best testing accuracy and also learns faster than the uncompressed algorithms.

Turning to the neural network experiment (cf. Figure 2), in terms of the final training gradient norm, BEER converges to a solution comparable to $D^2$, but at a lower communication cost, while CHOCO-SGD and DSGD cannot converge due to the data heterogeneity. In terms of testing accuracy, BEER and $D^2$ have very similar performance, and outperform CHOCO-SGD and DSGD.

**Convolutional neural network training**  We further compare the performance of BEER and CHOCO-SGD on training a convolutional neural network using the unshuffled `MNIST` dataset. Specifically, the network is consist of three modules: the first module is a 2-d convolution layer (1 input channel, 16 output channels, kernel size 5, stride 1 and padding 2) followed by 2-d batch normalization, ReLU activation and 2-d max pooling (kernel size 2 and stride 2); the second module is the same as the first module, except the convolution layer has 16 input channels and 32 output channels; the last module is a fully-connected layer with 1568 inputs and 10 outputs. We adopt the standard

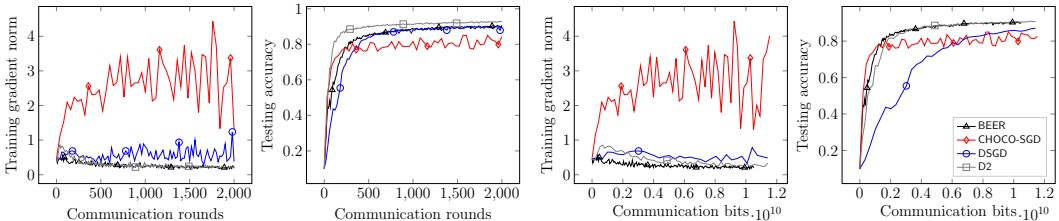

Figure 2: The training gradient norm and testing accuracy against communication rounds (left two panels) and communication bits (right two panels) for classification on unshuffled `MNIST` dataset using a 1-hidden-layer neural network. Both BEER and CHOCO-SGD employ the biased gsgd$_b$ compression [1] with $b = 20$.

cross-entropy loss, and simply average each agent's model with its neighbors. Figure 3 shows the testing gradient norm and accuracy against the communication bits. It can be seen that BEER outperforms CHOCO-SGD in terms of both testing gradient norm and testing accuracy. Both algorithms converge fast initially, however, due to to the extreme data heterogeneity, their convergence speeds significantly degenerate after a short time. BEER keeps improving the objective when CHOCO-SGD hits its error floor, which highlights BEER's advantage to deal with data heterogeneity.

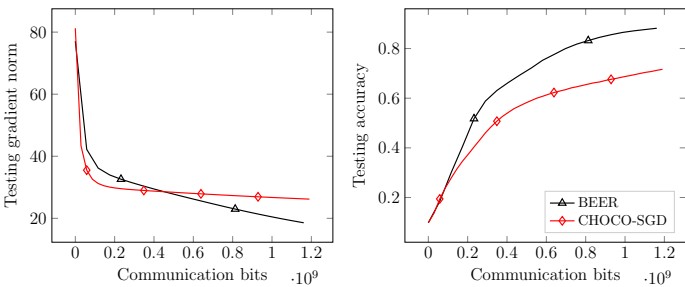

Figure 3: The testing gradient norm and testing accuracy against communication bits on unshuffled `MNIST` dataset using a 3-layer convolutional neural network. Both BEER and CHOCO-SGD employ the biased gsgd$_b$ compression [1] with $b = 5$.

In summary, BEER has much better performance in terms of communication efficiency than CHOCO-SGD in heterogeneous data scenario, which corroborates our theory. BEER also performs similarly or even better than the uncompressed baseline algorithm $D^2$, and much better than DSGD. In addition, by leveraging different communication compression schemes, BEER allows more flexible trade-offs between communication and computation, making it an appealing choice in practice.

## 6 Conclusion

This paper presents BEER, which achieves a faster $O(1/T)$ convergence rate for decentralized nonconvex optimization with communication compression, *without* imposing the bounded dissimilarity or bounded gradient assumptions. In addition, a faster linear rate of convergence is established for BEER under the PL condition. Numerical experiments are provided to corroborate our theory on the advantage of BEER in the data heterogeneous scenario. An interesting direction of future work is to investigate the lower bounds for decentralized (nonconvex) optimization with communication compression. In addition, improving the dependency of BEER with the network topology parameter $\rho$, possibly leveraging the analysis in Koloskova et al. [18], is of interest.

## Acknowledgement

The work of H. Zhao is supported in part by NSF, ONR, Simons Foundation, DARPA and SRC through awards to S. Arora. The work of B. Li, Z. Li and Y. Chi is supported in part by ONR N00014-19-1-2404, by AFRL under FA8750-20-2-0504, and by NSF under CCF-1901199, CCF-2007911 and CNS-2148212. B. Li is also gratefully supported by Wei Shen and Xuehong Zhang Presidential Fellowship at Carnegie Mellon University. The work of P. Richtárik is supported by KAUST Baseline Research Fund.

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
