# Appendix

## A  Related works

In this section, we review closely related literature on decentralized optimization, communication-efficient algorithms, and communication compression.

**Decentralized optimization** Decentralized optimization, which is a special class of linearly constrained (consensus constraint) optimization problems, has been studied for a long time [3, 9]. Many centralized algorithms can be intuitively converted into decentralized counterparts by using gossip averaging [14, 53], which mixes parameters from neighboring clients to enforce consensus.

However, direct applications of gossip averaging often lead to either slow convergence or high error floors [34], and many fixes have been proposed in response [44, 56, 38, 7, 35]. Among them, gradient tracking [38, 7, 35], which applies the idea of dynamic average consensus [59] to global gradient estimation, provides a systematic approach to reduce the variance and has been successfully applied to decentralize many algorithms with faster rates of convergence. For nonconvex problems, a small sample of gradient tracking aided algorithms include GT-SAGA [55], D-GET [48], GT-SARAH [54], and DESTRESS [22]. Our BEER algorithm also leverages gradient tracking to eliminate the strong bounded gradient and bounded dissimilarity assumptions.

**Communication-efficient algorithms** While decentralized optimization is a classical topic, the focus on communication efficiency is relatively new due to the advances in large-scale machine learning. Roughly speaking, there are primarily two kinds of approaches to reduce communication cost: 1) *local methods*: in each communication round, clients run multiple local update steps before communicating, in the hope of reducing the number of communication rounds; 2) *compressed methods*: clients send compressed communication messages, in the hope of reducing the communication cost per communication round.

Both categories have received significant attention in recent years. For local methods, a small sample of examples include FedAvg [32], Local-SVRG [10], SCAFFOLD [13] and FedPAGE [58]. On the other hand, many compressed methods are proposed recently such as [1, 15, 49, 47, 16, 26, 11, 25, 40, 8, 57]. In this paper, we will adopt the second approach based on communication compression to enhance communication efficiency.

**Decentralized nonconvex optimization with compression** As discussed earlier and summarized in Table 1, there have been limited existing works on decentralized nonconvex optimization with communication compression. In particular, SQuARM-SGD [45] can be viewed as CHOCO-SGD with momentum, but its theoretical convergence rate is slower than the original CHOCO-SGD. Deepsqueeze [51] and CHOCO-SGD have a close relationship, where Deepsqueeze can be viewed as decentralized SGD (DSGD or D-PSGD) with the explicit error feedback framework, and CHOCO-SGD uses control variables to implicitly handle the compression error.

## B  Examples of Compression Operators

We provide some examples of compression operators satisfying Definition 1 that are used in our experiments.

**gsgd$_b$ [1]**  gsgd$_b : \mathbb{R}^d \to \mathbb{R}^d$ $(b > 1)$, or random dithering, is a compression operator satisfying the following formula

$$\text{gsgd}_b(\boldsymbol{x}) := \frac{\|\boldsymbol{x}\|}{\tau} \cdot \text{sign}(\boldsymbol{x}) \cdot 2^{-(b-1)} \cdot \left\lfloor \frac{2^{(b-1)}|\boldsymbol{x}|}{\|\boldsymbol{x}\|} + \boldsymbol{u} \right\rfloor,$$

where $\tau = 1 + \min\left\{\frac{d}{2^{2(b-1)}}, \frac{\sqrt{d}}{2^{(b-1)}}\right\}$, and $\boldsymbol{u}$ is the random dithering vector uniformly sampled from $[0, 1]^d$. It follows that gsgd$_b$ satisfies Definition 1 with $\alpha = 1/\tau$.

**top$_k$ [2, 47]**  top$_k : \mathbb{R}^d \to \mathbb{R}^d$ is a compression operator satisfying the following formula

$$\text{top}_k(\boldsymbol{x}) := \boldsymbol{x} \odot \boldsymbol{u}(\boldsymbol{x}),$$

where $\boldsymbol{u}(\boldsymbol{x}) \in \{0,1\}^d$ that satisfies $\|\boldsymbol{u}\|_1 = k$ and $\boldsymbol{u}_i = 1$ for all $i \in \mathcal{I}$ such that $|\boldsymbol{x}_i| \geq |\boldsymbol{x}_j|$ for any $i \in \mathcal{I}$ and $j \in [d] \setminus \mathcal{I}$. In words, $\text{top}_k$ keeps the coordinates of $\boldsymbol{x}$ with the $k$ largest absolute values, and sets the other coordinates to 0. It follows that $\text{top}_k(\boldsymbol{x})$ satisfies Definition 1 with $\alpha = k/d$.

## C    Additional Experiments

This section provides two numerical experiments in addition to Section 5 to investigate BEER's performance under different network topologies and compression operators. Analogous to Section 5, we run logistic regression with nonconvex regularization ($\alpha = 0.05$) on unshuffled a9a dataset split evenly to 40 agents, and use FDLA matrix for weighted information aggregation.

### C.1    Comparison between different communication topologies

This experiment compares BEER's performance on the following network topologies: ring topology ($\rho = 0.022$), star topology ($\rho = 0.049$), grid topology ($\rho = 0.063$), and Erdös-Rènyi topology with connectivity probability $p = 0.5$ and $p = 0.2$ ($\rho = 0.51$ and $\rho = 0.77$, respectively). All experiments use the same best-tuned step size $\eta = 0.5$, batch size 100 and $\gamma = 0.7$, to guarantee convergence while achieving fast convergence.

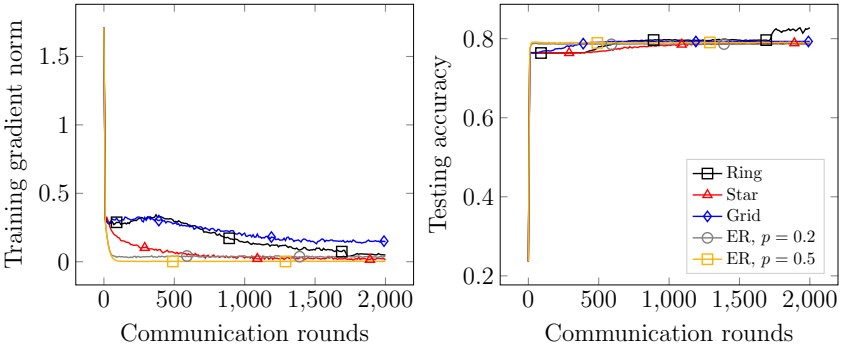

Figure 4: The training gradient norm and testing accuracy against communication rounds for BEER using biased $\text{gsgd}_b$ compression [1] with $b = 5$ on logistic regression with nonconvex regularization on unshuffled a9a dataset.

Figure 4 shows the experiment results, which empirically verifies that BEER is robust to network topologies, i.e., despite the huge difference of spectral gap $\rho$ between different topologies, BEER can use nearly the same hyperparameters to obtain similar performance. The experiment results complement our theoretical analysis and show that BEER may converge way better in practice despite its cubic dependency of $1/\rho$ in Theorem 2.

### C.2    Comparison between different compression schemes

This experiment compares BEER's performances on a ring topology using the following compression algorithms: identity (no compression), $\text{gsgd}_5$ and $\text{top}_{10}$, which are defined in Appendix B. Parameters are chosen such that BEER with different compression operators transfer similar amount of data per communication round. All experiments use the same best-tuned step size $\eta = 0.5$, batch size 100 and $\gamma = 0.7$, except that we use $\eta = 0.005$ and $\gamma = 0.8$ for $\text{top}_{10}$ compression.

Results are shown in Figure 5. From the right two panels in Figure 5, we can conclude using any compression operator outperforms the non-compressed baseline, in the sense that, all compressors converge to a solution with lower gradient norm and higher testing accuracy at a lower communication cost. From the second to the left panel in Figure 5, we can find in terms of communication rounds and testing accuracy, different compression operators can lead to significantly behaviors. For example, BEER with $\text{gsgd}_5$ compression operator converges faster than BEER without compression, but BEER with $\text{top}_{10}$ compression operator converges slower than BEER without compression. Among all experiments, BEER with $\text{gsgd}_5$ reaches the highest final testing accuracy, while behaves similar to

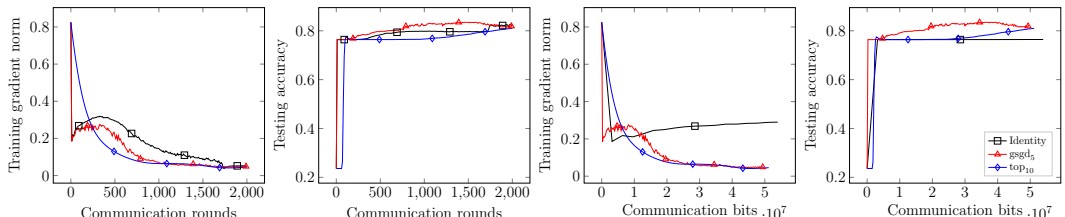

Figure 5: The training gradient norm and testing accuracy against communication rounds (left two panels) and communication bits (right two panels) for BEER using different compression algorithms on logistic regression with nonconvex regularization on unshuffled a9a dataset.

BEER without compression in terms of communication rounds, which implies that gsgd$_b$ (random quantization) may be a more practical compression operator, or may be more suitable for BEER.

# D    Proof of Main Theorems

## D.1    Technical preparation

We first recall some classical inequalities that helps our derivation.

**Proposition 1** *Let $\{\boldsymbol{v}_1, \ldots, \boldsymbol{v}_\tau\}$ be a set of $\tau$ vectors in $\mathbb{R}^d$. Then, $\forall \beta > 0$, we have*

$$\langle \boldsymbol{v}_i, \boldsymbol{v}_j \rangle \leq \frac{\beta}{2}\|\boldsymbol{u}\|^2 + \frac{1}{2\beta}\|\boldsymbol{v}\|^2, \tag{7}$$

$$\|\boldsymbol{v}_i + \boldsymbol{v}_j\|^2 \leq (1+\beta)\|\boldsymbol{v}_i\|^2 + \left(1 + \frac{1}{\beta}\right)\|\boldsymbol{v}_j\|^2, \tag{8}$$

$$\left\|\sum_{i=1}^{\tau} \boldsymbol{v}_i\right\|^2 \leq \tau \sum_{i=1}^{\tau} \|\boldsymbol{v}_i\|^2. \tag{9}$$

*Here, (7) is referred as the Cauchy-Schwarz inequality, (8) and (9) are referred as Young's inequality.*

**Additional notation**    The following notation will be used throughout our proof:
$$\nabla F(\boldsymbol{X}) := [\nabla f_1(\boldsymbol{x}_1), \nabla f_2(\boldsymbol{x}_2), \ldots, \nabla f_n(\boldsymbol{x}_n)],$$
$$\nabla F_b(\boldsymbol{X}) := [\nabla f_1(\boldsymbol{x}_1), \nabla f_2(\boldsymbol{x}_2), \ldots, \nabla f_n(\boldsymbol{x}_n)],$$
$$\nabla F(\bar{\boldsymbol{x}}) := [\nabla f_1(\bar{\boldsymbol{x}}), \nabla f_2(\bar{\boldsymbol{x}}), \ldots, \nabla f_n(\bar{\boldsymbol{x}})],$$
$$\tilde{\nabla}_b F(\bar{\boldsymbol{x}}) := [\tilde{\nabla}_b f_1(\bar{\boldsymbol{x}}), \tilde{\nabla}_b f_2(\bar{\boldsymbol{x}}), \ldots, \tilde{\nabla}_b f_n(\bar{\boldsymbol{x}})],$$
where $\bar{\boldsymbol{x}} := \frac{1}{n}\boldsymbol{X}\mathbf{1}$ with $\boldsymbol{X} = [\boldsymbol{x}_1, \boldsymbol{x}_2, \ldots, \boldsymbol{x}_n]$.

**Properties of the mixing matrix**    We make note of several useful properties of the mixing matrix in the following lemma.

**Lemma 1** *Let $\boldsymbol{W}$ be a mixing matrix satisfying Assumption 1 and has spectral gap $\rho$, then for any matrix $\boldsymbol{M} \in \mathbb{R}^{d \times n}$ and $\bar{\boldsymbol{m}} = \frac{1}{n}\boldsymbol{M}\mathbf{1}$, we have*

$$\left\|\boldsymbol{M}\boldsymbol{W} - \bar{\boldsymbol{m}}\mathbf{1}^\top\right\|_F^2 = \left\|\boldsymbol{M}\boldsymbol{W} - \bar{\boldsymbol{m}}\mathbf{1}^\top \boldsymbol{W}\right\|_F^2 \leq (1-\rho)\left\|\boldsymbol{M} - \bar{\boldsymbol{m}}\mathbf{1}^\top\right\|_F^2. \tag{10}$$

*In addition, for any $\gamma \in (0, 1]$, the matrix $\widetilde{\boldsymbol{W}} = \boldsymbol{I} + \gamma(\boldsymbol{W} - \boldsymbol{I})$ satisfies Assumption 1 with a spectral gap at least $\gamma\rho$.*

*Proof:* The first claim follows from the spectral decomposition of $\boldsymbol{W}$. Since $\boldsymbol{W}$ is a doubly stochastic matrix, the largest absolute eigenvalue of $\boldsymbol{W}$ is 1 and the corresponding eigenvector is $\mathbf{1}$. Let $\boldsymbol{v}_2, \ldots, \boldsymbol{v}_n$ be the eigenvectors of $\boldsymbol{W}$ corresponding to the remaining eigenvalues. Then, we have

$$\left\|\boldsymbol{M}\boldsymbol{W} - \bar{\boldsymbol{m}}\mathbf{1}^\top\right\|_F^2 = \left\|\boldsymbol{M}\boldsymbol{W} - \bar{\boldsymbol{m}}\mathbf{1}^\top \boldsymbol{W}\right\|_F^2 = \sum_{i=1}^{r}\|\boldsymbol{W}(\boldsymbol{m}_i - \bar{\boldsymbol{m}}_i\mathbf{1})\|^2,$$

where the first equality follows from $\mathbf{1}^\top \mathbf{W} = \mathbf{1}^\top$, $\mathbf{m}_i$ denotes the transpose of $i$-th row of matrix $\mathbf{M}$, and $\bar{m}_i$ denotes the average of $\mathbf{m}_i$. Now we decompose $\mathbf{m}_i - \bar{m}_i \mathbf{1}$ using the eigenvectors of $\mathbf{W}$. Noting that

$$\mathbf{1}^\top (\mathbf{m}_i - \bar{m}_i \mathbf{1}) = \mathbf{1}^\top \mathbf{m}_i - \mathbf{1}^\top \mathbf{1} \frac{1}{n} \mathbf{1}^\top \mathbf{m}_i = 0,$$

and thus we can write

$$\mathbf{m}_i - \bar{m}_i \mathbf{1} = \sum_{j=2}^{n} c_j \mathbf{v}_j$$

for some $\{c_j\}_{j=2}^n$. Then, we have

$$\|\mathbf{W}(\mathbf{m}_i - \bar{m}_i\mathbf{1})\|^2 = \left\| \mathbf{W} \sum_{j=2}^{n} c_j \mathbf{v}_j \right\|^2 \leq (1-\rho)^2 \sum_{j=2}^{n} c_j^2 \leq (1-\rho) \sum_{j=2}^{n} c_j^2 = (1-\rho) \|\mathbf{m}_i - \bar{m}_i\mathbf{1}\|^2,$$

and we conclude the proof of this claim.

For the second claim, recall the fact that if $\mathbf{v}$ is an eigenvector of $\mathbf{W}$ corresponding to the eigenvalue $\lambda$, then $\mathbf{v}$ is also an eigenvector of $\widetilde{\mathbf{W}}$ with the corresponding eigenvalue $(1 - \gamma) + \gamma\lambda$. This claim follows from simple computation based on this relation. $\square$

**A key consequence of gradient tracking** Before diving in the proofs of the main theorems, we record a key property of gradient tracking. Specifically, we have the following lemma.

**Lemma 2** *If* $\bar{v}^0 = \frac{1}{n}\tilde{\nabla}_b F(\mathbf{X}^0)\mathbf{1}$, *then for any* $t \geq 1$, *we have*

$$\bar{v}^t = \frac{1}{n}\tilde{\nabla}_b F(\mathbf{X}^t)\mathbf{1}, \tag{11}$$

*and*

$$\bar{x}^{t+1} = \bar{x}^t - \frac{\eta}{n}\tilde{\nabla}_b F(\mathbf{X}^t)\mathbf{1}. \tag{12}$$

**Proof:** We first prove (11) by induction. For the base case ($t = 0$), the relation (11) is obviously true by the means of initialization. Now suppose that at the $t$-th iteration, the relation (11) is true, i.e.,

$$\bar{v}^t = \frac{1}{n}\tilde{\nabla}_b F(\mathbf{X}^t)\mathbf{1},$$

then at the $(t + 1)$-th iteration, we have

$$\begin{aligned}
\bar{v}^{t+1} &= \frac{1}{n}\mathbf{V}^{t+1}\mathbf{1} \\
&= \frac{1}{n}\mathbf{V}^t\mathbf{1} + \frac{1}{n}\gamma\mathbf{G}^t(\mathbf{W} - \mathbf{I})\mathbf{1} + \frac{1}{n}\left(\tilde{\nabla}_b F(\mathbf{X}^{t+1}) - \tilde{\nabla}_b F(\mathbf{X}^t)\right)\mathbf{1} \\
&= \frac{1}{n}\mathbf{V}^t\mathbf{1} + \frac{1}{n}\left(\tilde{\nabla}_b F(\mathbf{X}^{t+1}) - \tilde{\nabla}_b F(\mathbf{X}^t)\right)\mathbf{1} \\
&= \frac{1}{n}\tilde{\nabla}_b F(\mathbf{X}^{t+1})\mathbf{1}.
\end{aligned} \tag{13}$$

where (13) follows from the update rule of BEER (cf. Line 6), the penultimate line follows from $\mathbf{W}\mathbf{1} = \mathbf{1}$, and the last line follows from the induction hypothesis at the $t$-th iteration. Thus the induction hypothesis is also true at the $(t + 1)$-th iteration, and we complete the proof of (11).

For (12), it follows from the update rule of BEER (cf. Line 3) that

$$\begin{aligned}
\bar{x}^{t+1} &= \bar{x}^t + \frac{\gamma}{n}\mathbf{H}^t(\mathbf{W} - \mathbf{I})\mathbf{1} - \frac{\eta}{n}\mathbf{V}^t\mathbf{1} \\
&= \bar{x}^t - \eta\bar{v}^t = \bar{x}^t - \frac{\eta}{n}\tilde{\nabla}_b F(\mathbf{X}^t)\mathbf{1},
\end{aligned}$$

where the second line uses $\mathbf{W}\mathbf{1} = \mathbf{1}$ and (11). $\square$

## D.2 Recursive relations of main errors

As mentioned previously, the proof is centered around controlling the following set of errors which we repeat below for convenience (cf. (5)),

(compression approximation error:) $\quad \Omega_1^t = \mathbb{E} \left\| \boldsymbol{H}^t - \boldsymbol{X}^t \right\|_{\mathrm{F}}^2, \qquad \Omega_2^t = \mathbb{E} \left\| \boldsymbol{G}^t - \boldsymbol{V}^t \right\|_{\mathrm{F}}^2,$

(consensus error:) $\quad \Omega_3^t = \mathbb{E} \left\| \boldsymbol{X}^t - \bar{\boldsymbol{x}}^t \mathbf{1}^\top \right\|_{\mathrm{F}}^2, \qquad \Omega_4^t = \mathbb{E} \left\| \boldsymbol{V}^t - \bar{\boldsymbol{v}}^t \mathbf{1}^\top \right\|_{\mathrm{F}}^2,$

(gradient norm:) $\quad \Omega_5^t = \mathbb{E} \left\| \bar{\boldsymbol{v}}^t \right\|^2.$

In particular, we aim to build a set of recursive relations of $\Omega_1^t$ to $\Omega_4^t$, which will be specified in the following lemma.

**Lemma 3** *Suppose Assumptions 1, 2 and 3 hold, then for any $t \geq 0$, we have*

$$\Omega_1^{t+1} \leq \left(1 - \frac{\alpha}{2} + \frac{6\gamma^2 C}{\alpha}\right) \Omega_1^t + 0 \cdot \Omega_2^t + \frac{6\gamma^2 C}{\alpha} \Omega_3^t + \frac{6\eta^2}{\alpha} \Omega_4^t + \frac{6n\eta^2}{\alpha} \Omega_5^t, \tag{14a}$$

$$\Omega_2^{t+1} \leq \frac{18\gamma^2 C L^2}{\alpha} \Omega_1^t + \left(1 - \frac{\alpha}{2} + \frac{6\gamma^2 C}{\alpha}\right) \Omega_2^t + \frac{18\gamma^2 C L^2}{\alpha} \Omega_3^t$$
$$+ \frac{6\gamma^2 C + 18 L^2 \eta^2}{\alpha} \Omega_4^t + \frac{18 L^2 \eta^2 n}{\alpha} \Omega_5^t + \frac{12 n \sigma^2}{b\alpha}, \tag{14b}$$

$$\Omega_3^{t+1} \leq \frac{6\gamma C}{\rho} \Omega_1^t + 0 \cdot \Omega_2^t + \left(1 - \frac{\gamma\rho}{2}\right) \Omega_3^t + \frac{6\eta^2}{\gamma\rho} \Omega_4^t + 0 \cdot \Omega_5^t, \tag{14c}$$

$$\Omega_4^{t+1} \leq \frac{18\gamma C L^2}{\rho} \Omega_1^t + \frac{6\gamma C}{\rho} \Omega_2^t + \frac{18\gamma C L^2}{\rho} \Omega_3^t$$
$$+ \left(1 - \frac{\gamma\rho}{2} + \frac{18 L^2 \eta^2}{\gamma\rho}\right) \Omega_4^t + \frac{18 n\eta^2 L^2}{\gamma\rho} \cdot \Omega_5^t + \frac{12 n \sigma^2}{b\gamma\rho}, \tag{14d}$$

*where*

$$C = \|\boldsymbol{W} - \boldsymbol{I}\|^2 = \sigma_{\max}(\boldsymbol{W} - \boldsymbol{I})^2 \tag{15}$$

*is the square of the maximum singular value of the matrix $\boldsymbol{W} - \boldsymbol{I}$.*

Note that the eigenvalues of $\boldsymbol{W}$ and $\boldsymbol{I}$ all lies in $[-1, 1]$, and thus clearly $C \leq 4$.

**Proof:** We will establish the inequalities in (14) one by one.

**Bounding $\Omega_1^t$ in (14a)** First from the update rule of BEER (cf. Line 5), we have

$$\left\| \boldsymbol{H}^{t+1} - \boldsymbol{X}^{t+1} \right\|_{\mathrm{F}}^2 = \left\| \boldsymbol{H}^t + \mathcal{C}(\boldsymbol{X}^{t+1} - \boldsymbol{H}^t) - \boldsymbol{X}^{t+1} \right\|_{\mathrm{F}}^2$$
$$\leq (1 - \alpha) \left\| \boldsymbol{X}^{t+1} - \boldsymbol{H}^t \right\|_{\mathrm{F}}^2$$
$$\leq \left(1 - \frac{\alpha}{2}\right) \left\| \boldsymbol{X}^t - \boldsymbol{H}^t \right\|_{\mathrm{F}}^2 + \frac{2}{\alpha} \left\| \boldsymbol{X}^{t+1} - \boldsymbol{X}^t \right\|_{\mathrm{F}}^2, \tag{16}$$

where the first inequality comes from the definition of compression operators (Definition 1) and the second inequality comes from Young's inequality. It then boils down to bound $\left\| \boldsymbol{X}^{t+1} - \boldsymbol{X}^t \right\|_{\mathrm{F}}^2$, for which we have

$$\left\| \boldsymbol{X}^{t+1} - \boldsymbol{X}^t \right\|_{\mathrm{F}}^2$$
$$= \left\| \gamma \boldsymbol{H}^t (\boldsymbol{W} - \boldsymbol{I}) - \eta \boldsymbol{V}^t \right\|_{\mathrm{F}}^2$$
$$= \left\| \gamma (\boldsymbol{H}^t - \boldsymbol{X}^t)(\boldsymbol{W} - \boldsymbol{I}) + \gamma (\boldsymbol{X}^t - \bar{\boldsymbol{x}}^t \mathbf{1}^\top)(\boldsymbol{W} - \boldsymbol{I}) - \eta \boldsymbol{V}^t \right\|_{\mathrm{F}}^2$$
$$\leq 3\gamma^2 C \left\| \boldsymbol{X}^t - \boldsymbol{H}^t \right\|_{\mathrm{F}} + 3\gamma^2 C \left\| \boldsymbol{X}^t - \bar{\boldsymbol{x}}^t \mathbf{1}^\top \right\|_{\mathrm{F}}^2 + 3\eta^2 \left\| \boldsymbol{V}^t \right\|_{\mathrm{F}}^2$$
$$= 3\gamma^2 C \left\| \boldsymbol{X}^t - \boldsymbol{H}^t \right\|_{\mathrm{F}} + 3\gamma^2 C \left\| \boldsymbol{X}^t - \bar{\boldsymbol{x}}^t \mathbf{1}^\top \right\|_{\mathrm{F}}^2 + 3\eta^2 \left\| \boldsymbol{V}^t - \bar{\boldsymbol{v}}^t \mathbf{1}^\top \right\|_{\mathrm{F}}^2 + 3\eta^2 n \left\| \bar{\boldsymbol{v}}^t \right\|^2, \tag{17}$$

where in the first line we use the update rule of BEER (cf. Line 3), in the second line we use the property of the mixing matrix $\mathbf{1}^\top \boldsymbol{W} = \mathbf{1}^\top$, and in the third line, we apply Young's inequality (cf. (9)).

In the fourth line, we use $\|\boldsymbol{v}\|^2 = \|\boldsymbol{v} - \bar{v}\mathbf{1}\|^2 + n\bar{v}^2$ for any vector $\boldsymbol{v}$ with an average $\bar{v}$. Plugging this back into (16), we get

$$\left\|\boldsymbol{H}^{t+1} - \boldsymbol{X}^{t+1}\right\|_{\mathrm{F}}^2 \leq \left(1 - \frac{\alpha}{2} + \frac{6\gamma^2 C}{\alpha}\right)\left\|\boldsymbol{X}^t - \boldsymbol{H}^t\right\|_{\mathrm{F}}^2 + \frac{6\gamma^2 C}{\alpha}\left\|\boldsymbol{X}^t - \bar{\boldsymbol{x}}^t\mathbf{1}^\top\right\|_{\mathrm{F}}^2$$

$$+ \frac{6\eta^2}{\alpha}\left\|\boldsymbol{V}^t - \bar{\boldsymbol{v}}^t\mathbf{1}^\top\right\|_{\mathrm{F}}^2 + \frac{6n\eta^2}{\alpha}\left\|\bar{\boldsymbol{v}}^t\right\|^2 .$$

Plugging in the definitions of $\Omega_i^t$, we obtain (14a).

**Bounding $\Omega_2^t$ in (14b)** Similar to the derivation of (14a), by applying the update rule of $\boldsymbol{G}^t$ in BEER (Line 8), the definition of compression operators (Definition 1), and Young's inequality, we have

$$\left\|\boldsymbol{V}^{t+1} - \boldsymbol{G}^{t+1}\right\|_{\mathrm{F}}^2 = \left\|\boldsymbol{G}^t + \mathcal{C}(\boldsymbol{V}^{t+1} - \boldsymbol{G}^t) - \boldsymbol{V}^{t+1}\right\|_{\mathrm{F}}^2$$

$$\leq (1-\alpha)\left\|\boldsymbol{G}^t - \boldsymbol{V}^{t+1}\right\|_{\mathrm{F}}^2$$

$$\leq \left(1 - \frac{\alpha}{2}\right)\left\|\boldsymbol{G}^t - \boldsymbol{V}^t\right\|_{\mathrm{F}}^2 + \frac{2}{\alpha}\left\|\boldsymbol{V}^{t+1} - \boldsymbol{V}^t\right\|_{\mathrm{F}}^2 . \qquad (18)$$

It then boils down to bound $\left\|\boldsymbol{V}^{t+1} - \boldsymbol{V}^t\right\|_{\mathrm{F}}^2$. By the update rule of BEER (cf. Line 6), we have

$$\left\|\boldsymbol{V}^{t+1} - \boldsymbol{V}^t\right\|_{\mathrm{F}}^2$$

$$= \left\|\gamma\boldsymbol{G}^t(\boldsymbol{W} - \boldsymbol{I}) + (\tilde{\nabla}_b F(\boldsymbol{X}^{t+1}) - \tilde{\nabla}_b F(\boldsymbol{X}^t))\right\|_{\mathrm{F}}^2$$

$$= \left\|\gamma(\boldsymbol{G}^t - \boldsymbol{V}^t)(\boldsymbol{W} - \boldsymbol{I}) + \gamma(\boldsymbol{V}^t - \bar{\boldsymbol{v}}^t\mathbf{1}^\top)(\boldsymbol{W} - \boldsymbol{I}) + (\tilde{\nabla}_b F(\boldsymbol{X}^{t+1}) - \tilde{\nabla}_b F(\boldsymbol{X}^t))\right\|_{\mathrm{F}}^2$$

$$\overset{(i)}{\leq} 3\gamma^2 C\left\|\boldsymbol{G}^t - \boldsymbol{V}^t\right\|_{\mathrm{F}}^2 + 3\gamma^2 C\left\|\boldsymbol{V}^t - \bar{\boldsymbol{v}}^t\mathbf{1}^\top\right\|_{\mathrm{F}}^2 + 3\left\|\tilde{\nabla}_b F(\boldsymbol{X}^{t+1}) - \tilde{\nabla}_b F(\boldsymbol{X}^t)\right\|_{\mathrm{F}}^2$$

$$\overset{(ii)}{\leq} 3\gamma^2 C\left\|\boldsymbol{G}^t - \boldsymbol{V}^t\right\|_{\mathrm{F}}^2 + 3\gamma^2 C\left\|\boldsymbol{V}^t - \bar{\boldsymbol{v}}^t\mathbf{1}^\top\right\|_{\mathrm{F}}^2 + 3\left\|\nabla F(\boldsymbol{X}^{t+1}) - \nabla F(\boldsymbol{X}^t)\right\|_{\mathrm{F}}^2 + \frac{6n\sigma^2}{b}$$

$$\overset{(iii)}{\leq} 3\gamma^2 C\left\|\boldsymbol{G}^t - \boldsymbol{V}^t\right\|_{\mathrm{F}}^2 + 3\gamma^2 C\left\|\boldsymbol{V}^t - \bar{\boldsymbol{v}}^t\mathbf{1}^\top\right\|_{\mathrm{F}}^2 + 3L^2\left\|\boldsymbol{X}^{t+1} - \boldsymbol{X}^t\right\|_{\mathrm{F}}^2 + \frac{6n\sigma^2}{b}$$

$$\overset{(iv)}{\leq} 3\gamma^2 C\left\|\boldsymbol{G}^t - \boldsymbol{V}^t\right\|_{\mathrm{F}}^2 + (3\gamma^2 C + 9L^2\eta^2)\left\|\boldsymbol{V}^t - \bar{\boldsymbol{v}}^t\mathbf{1}^\top\right\|_{\mathrm{F}}^2$$

$$+ 9\gamma^2 CL^2\left\|\boldsymbol{X}^t - \boldsymbol{H}^t\right\|_{\mathrm{F}}^2 + 9\gamma^2 CL^2\left\|\boldsymbol{X}^t - \bar{\boldsymbol{x}}^t\mathbf{1}^\top\right\|_{\mathrm{F}}^2 + 9L^2\eta^2 n\left\|\bar{\boldsymbol{v}}^t\right\|^2 + \frac{6n\sigma^2}{b},$$

where (i) comes from Young's inequality (cf. (9)) and basic facts of matrix norm (cf. (15)), (ii) comes from the bounded variance assumption (Assumption 3), (iii) comes from the smoothness assumption (Assumption 2), and (iv) follows from (17). Combining the above inequality with (18), we have

$$\left\|\boldsymbol{V}^{t+1} - \boldsymbol{G}^{t+1}\right\|_{\mathrm{F}}^2$$

$$\leq \left(1 - \frac{\alpha}{2}\right)\left\|\boldsymbol{G}^t - \boldsymbol{V}^t\right\|_{\mathrm{F}}^2 + \frac{2}{\alpha}\left\|\boldsymbol{V}^{t+1} - \boldsymbol{V}^t\right\|_{\mathrm{F}}^2$$

$$\leq \left(1 - \frac{\alpha}{2} + \frac{6\gamma^2 C}{\alpha}\right)\left\|\boldsymbol{G}^t - \boldsymbol{V}^t\right\|_{\mathrm{F}}^2 + \frac{6\gamma^2 C + 18L^2\eta^2}{\alpha}\left\|\boldsymbol{V}^t - \bar{\boldsymbol{v}}^t\mathbf{1}^\top\right\|_{\mathrm{F}}^2$$

$$+ \frac{18\gamma^2 CL^2}{\alpha}\left\|\boldsymbol{X}^t - \boldsymbol{H}^t\right\|_{\mathrm{F}}^2 + \frac{18\gamma^2 CL^2}{\alpha}\left\|\boldsymbol{X}^t - \bar{\boldsymbol{x}}^t\mathbf{1}^\top\right\|_{\mathrm{F}}^2 + \frac{18L^2\eta^2 n}{\alpha}\left\|\bar{\boldsymbol{v}}^t\right\|^2 + \frac{12n\sigma^2}{b\alpha} .$$

Plugging in the definitions of $\Omega_i^t$, we obtain (14b).

**Bounding $\Omega_3^t$ in (14c)** To bound the consensus error $\left\|\boldsymbol{X}^{t+1} - \bar{\boldsymbol{x}}^{t+1}\mathbf{1}^\top\right\|_{\mathrm{F}}^2$, by the update rule of BEER (cf. Line 3), we have

$$\left\|\boldsymbol{X}^{t+1} - \bar{\boldsymbol{x}}^{t+1}\mathbf{1}^\top\right\|_{\mathrm{F}}^2$$

$$= \left\|\boldsymbol{X}^t + \gamma\boldsymbol{H}^t(\boldsymbol{W} - \boldsymbol{I}) - \eta\boldsymbol{V}^t - \bar{\boldsymbol{x}}^t\mathbf{1}^\top + \eta\bar{\boldsymbol{v}}^t\mathbf{1}^\top\right\|_{\mathrm{F}}^2$$

$$\overset{(i)}{=} \left\| \boldsymbol{X}^t \widetilde{\boldsymbol{W}} - \bar{\boldsymbol{x}}^t \mathbf{1}^\top + \gamma (\boldsymbol{H}^t - \boldsymbol{X}^t)(\boldsymbol{W} - \boldsymbol{I}) - \eta \boldsymbol{V}^t + \eta \bar{\boldsymbol{v}}^t \mathbf{1}^\top \right\|_\mathrm{F}^2$$

$$\overset{(ii)}{\le} (1+\beta)(1-\gamma\rho) \left\| \boldsymbol{X}^t - \bar{\boldsymbol{x}}^t \mathbf{1}^\top \right\|_\mathrm{F}^2 + \left(1 + \frac{1}{\beta}\right) \left( 2\gamma^2 \left\| (\boldsymbol{H}^t - \boldsymbol{X}^t)(\boldsymbol{W} - \boldsymbol{I}) \right\|_\mathrm{F}^2 + 2\eta^2 \left\| \boldsymbol{V}^t - \bar{\boldsymbol{v}}^t \mathbf{1}^\top \right\|_\mathrm{F}^2 \right)$$

$$\overset{(iii)}{\le} \left(1 - \frac{\gamma\rho}{2}\right) \left\| \boldsymbol{X}^t - \bar{\boldsymbol{x}}^t \mathbf{1}^\top \right\|_\mathrm{F}^2 + \left(1 + \frac{2}{\gamma\rho}\right) \left( 2\gamma^2 \left\| (\boldsymbol{H}^t - \boldsymbol{X}^t)(\boldsymbol{W} - \boldsymbol{I}) \right\|_\mathrm{F}^2 + 2\eta^2 \left\| \boldsymbol{V}^t - \bar{\boldsymbol{v}}^t \mathbf{1}^\top \right\|_\mathrm{F}^2 \right)$$

$$\overset{(iv)}{\le} \left(1 - \frac{\gamma\rho}{2}\right) \left\| \boldsymbol{X}^t - \bar{\boldsymbol{x}}^t \mathbf{1}^\top \right\|_\mathrm{F}^2 + \left(1 + \frac{2}{\gamma\rho}\right) \left( 2\gamma^2 C \left\| \boldsymbol{H}^t - \boldsymbol{X}^t \right\|_\mathrm{F}^2 + 2\eta^2 \left\| \boldsymbol{V}^t - \bar{\boldsymbol{v}}^t \mathbf{1}^\top \right\|_\mathrm{F}^2 \right)$$

$$\le \left(1 - \frac{\gamma\rho}{2}\right) \left\| \boldsymbol{X}^t - \bar{\boldsymbol{x}}^t \mathbf{1}^\top \right\|_\mathrm{F}^2 + \frac{6\gamma C}{\rho} \left\| \boldsymbol{H}^t - \boldsymbol{X}^t \right\|_\mathrm{F}^2 + \frac{6\eta^2}{\gamma\rho} \left\| \boldsymbol{V}^t - \bar{\boldsymbol{v}}^t \mathbf{1}^\top \right\|_\mathrm{F}^2,$$

where (i) follows from the definition $\widetilde{\boldsymbol{W}} = \boldsymbol{I} + \gamma(\boldsymbol{W} - \boldsymbol{I})$, (ii) follows from applying Young's inequality twice and Lemma 1, i.e.

$$\left\| \boldsymbol{X}^t \widetilde{\boldsymbol{W}} - \bar{\boldsymbol{x}}^t \mathbf{1}^\top \right\|_\mathrm{F} \le (1 - \gamma\rho) \left\| \boldsymbol{X}^t - \bar{\boldsymbol{x}}^t \mathbf{1}^\top \right\|_\mathrm{F}^2,$$

(iii) follows by choosing $\beta = \gamma\rho/2$, and (iv) uses the definition of $C$ (cf. (15)). Plugging in the definitions of $\Omega_i^t$, we obtain (14c).

**Bounding $\Omega_4^t$ in (14d)**   First, note that

$$\left\| \boldsymbol{V}^{t+1} - \bar{\boldsymbol{v}}^{t+1} \mathbf{1}^\top \right\|_\mathrm{F}^2 = \left\| \boldsymbol{V}^{t+1} - \bar{\boldsymbol{v}}^t \mathbf{1}^\top + \bar{\boldsymbol{v}}^t \mathbf{1}^\top - \bar{\boldsymbol{v}}^{t+1} \mathbf{1}^\top \right\|_\mathrm{F}^2$$
$$= \left\| \boldsymbol{V}^{t+1} - \bar{\boldsymbol{v}}^t \mathbf{1}^\top \right\|_\mathrm{F}^2 - n \left\| \bar{\boldsymbol{v}}^{t+1} - \bar{\boldsymbol{v}}^t \right\|^2$$
$$\le \left\| \boldsymbol{V}^{t+1} - \bar{\boldsymbol{v}}^t \mathbf{1}^\top \right\|_\mathrm{F}^2.$$

Thus by the update rule of BEER (cf. Line 6), we have

$$\left\| \boldsymbol{V}^{t+1} - \bar{\boldsymbol{v}}^{t+1} \mathbf{1}^\top \right\|_\mathrm{F}^2$$
$$\le \left\| \boldsymbol{V}^{t+1} - \bar{\boldsymbol{v}}^t \mathbf{1}^\top \right\|_\mathrm{F}^2$$
$$= \left\| \boldsymbol{V}^t + \gamma \boldsymbol{G}^{t+1}(\boldsymbol{W} - \boldsymbol{I}) + \tilde{\nabla}_b F(\boldsymbol{X}^{t+1}) - \tilde{\nabla}_b F(\boldsymbol{X}^t) - \bar{\boldsymbol{v}}^t \mathbf{1}^\top \right\|_\mathrm{F}^2$$
$$= \left\| (\boldsymbol{V}^t \widetilde{\boldsymbol{W}} - \bar{\boldsymbol{v}}^t \mathbf{1}^\top) + \gamma (\boldsymbol{G}^{t+1} - \boldsymbol{V}^t)(\boldsymbol{W} - \boldsymbol{I}) + (\tilde{\nabla}_b F(\boldsymbol{X}^{t+1}) - \tilde{\nabla}_b F(\boldsymbol{X}^t)) \right\|_\mathrm{F}^2$$
$$\overset{(i)}{\le} \left(1 - \frac{\gamma\rho}{2}\right) \left\| \boldsymbol{V}^t - \bar{\boldsymbol{v}}^t \mathbf{1}^\top \right\|_\mathrm{F}^2 + \left(1 + \frac{2}{\gamma\rho}\right) \left( 2\gamma^2 C \left\| \boldsymbol{G}^t - \boldsymbol{V}^t \right\|_\mathrm{F}^2 + 2L^2 \left\| \boldsymbol{X}^{t+1} - \boldsymbol{X}^t \right\|_\mathrm{F}^2 + \frac{4n\sigma^2}{b} \right)$$
$$\overset{(ii)}{\le} \left(1 - \frac{\gamma\rho}{2}\right) \left\| \boldsymbol{V}^t - \bar{\boldsymbol{v}}^t \mathbf{1}^\top \right\|_\mathrm{F}^2 + \frac{6\gamma C}{\rho} \left\| \boldsymbol{G}^t - \boldsymbol{V}^t \right\|_\mathrm{F}^2 + \frac{6L^2}{\gamma\rho} \left\| \boldsymbol{X}^{t+1} - \boldsymbol{X}^t \right\|_\mathrm{F}^2 + \frac{12n\sigma^2}{b\gamma\rho}$$
$$\le \left(1 - \frac{\gamma\rho}{2} + \frac{18L^2\eta^2}{\gamma\rho}\right) \left\| \boldsymbol{V}^t - \bar{\boldsymbol{v}}^t \mathbf{1}^\top \right\|_\mathrm{F}^2 + \frac{6\gamma C}{\rho} \left\| \boldsymbol{G}^t - \boldsymbol{V}^t \right\|_\mathrm{F}^2$$
$$+ \frac{18\gamma C L^2}{\rho} \left\| \boldsymbol{X}^t - \boldsymbol{H}^t \right\|_\mathrm{F}^2 + \frac{18\gamma C L^2}{\rho} \left\| \boldsymbol{X}^t - \bar{\boldsymbol{x}}^t \mathbf{1}^\top \right\|_\mathrm{F}^2 + \frac{18n\eta^2 L^2}{\gamma\rho} \left\| \bar{\boldsymbol{v}}^t \right\|^2 + \frac{12n\sigma^2}{b\gamma\rho},$$

where (i) and (ii) are obtained similarly as the derivation of (14c), and the last line follows from (17). Thus, we can get (14d) by plugging in the definitions of $\Omega_i^t$ and conclude the proof.   $\square$

### D.3   Proof of Theorem 1 and 2

Note that Theorem 2 is a strict generalization of Theorem 1, and thus we will directly prove Theorem 2. This proof makes use of Lemma 2 and Lemma 3, by constructing some proper Lyapunov function and demonstrate its descending property using a linear system argument, which is also used in, e.g., Li et al. [22], Liao et al. [29].

**Step 1: establishing a "descent" property of the function value** First, we have the following inequality captures the "descent" of the function value.

$f(\bar{\boldsymbol{x}}^{t+1})$

$\overset{(i)}{\leq} f(\bar{\boldsymbol{x}}^t) - \eta \langle \bar{\boldsymbol{v}}^t, \nabla f(\bar{\boldsymbol{x}}^t) \rangle + \frac{\eta^2 L}{2} \|\bar{\boldsymbol{v}}^t\|^2$

$= f(\bar{\boldsymbol{x}}^t) - \frac{\eta}{2} \|\nabla f(\bar{\boldsymbol{x}}^t)\|^2 - \frac{\eta}{2} \|\bar{\boldsymbol{v}}^t\|^2 + \frac{\eta}{2} \|\nabla f(\bar{\boldsymbol{x}}^t) - \bar{\boldsymbol{v}}^t\|^2 + \frac{\eta^2 L}{2} \|\bar{\boldsymbol{v}}^t\|^2$

$= f(\bar{\boldsymbol{x}}^t) - \frac{\eta}{2} \|\nabla f(\bar{\boldsymbol{x}}^t)\|^2 + \frac{\eta}{2} \|\nabla f(\bar{\boldsymbol{x}}^t) - \bar{\boldsymbol{v}}^t\|^2 - \left(\frac{\eta}{2} - \frac{\eta^2 L}{2}\right) \|\bar{\boldsymbol{v}}^t\|^2$

$\overset{(ii)}{\leq} f(\bar{\boldsymbol{x}}^t) - \frac{\eta}{2} \|\nabla f(\bar{\boldsymbol{x}}^t)\|^2 + \frac{\eta}{2n^2} \left\|\nabla F(\bar{\boldsymbol{x}}^t)\mathbf{1} - \tilde{\nabla}_b F(\boldsymbol{X}^t)\mathbf{1}\right\|^2 - \left(\frac{\eta}{2} - \frac{\eta^2 L}{2}\right) \|\bar{\boldsymbol{v}}^t\|^2$

$= f(\bar{\boldsymbol{x}}^t) - \frac{\eta}{2} \|\nabla f(\bar{\boldsymbol{x}}^t)\|^2 + \frac{\eta}{2n^2} \left\|\nabla F(\bar{\boldsymbol{x}}^t)\mathbf{1} - \nabla F(\boldsymbol{X}^t)\mathbf{1}\right\|^2 - \left(\frac{\eta}{2} - \frac{\eta^2 L}{2}\right) \|\bar{\boldsymbol{v}}^t\|^2$

$\quad + \frac{\eta}{2n^2} \left\|\nabla F(\boldsymbol{X}^t)\mathbf{1} - \tilde{\nabla}_b F(\boldsymbol{X}^t)\mathbf{1}\right\|^2 + \frac{\eta}{n^2} \left\langle \nabla F(\boldsymbol{X}^t)\mathbf{1} - \tilde{\nabla}_b F(\boldsymbol{X}^t)\mathbf{1}, \nabla F(\bar{\boldsymbol{x}}^t)\mathbf{1} - \nabla F(\boldsymbol{X}^t)\mathbf{1} \right\rangle,$

where (i) comes from the $L$-smooth assumption (Assumption 2), (ii) comes from Lemma 2. Take expectation on both sides, and using the bounded variance assumption (Assumption 3) and independence of stochastic samples, we get

$\mathbb{E}f(\bar{\boldsymbol{x}}^{t+1})$

$\leq \mathbb{E}f(\bar{\boldsymbol{x}}^t) - \frac{\eta}{2}\mathbb{E}\|\nabla f(\bar{\boldsymbol{x}}^t)\|^2 + \frac{\eta}{2n^2}\mathbb{E}\|\nabla F(\bar{\boldsymbol{x}}^t)\mathbf{1} - \nabla F(\boldsymbol{X}^t)\mathbf{1}\|^2 - \left(\frac{\eta}{2} - \frac{\eta^2 L}{2}\right)\mathbb{E}\|\bar{\boldsymbol{v}}^t\|^2 + \frac{\eta\sigma^2}{2bn}$

$\overset{(i)}{\leq} \mathbb{E}f(\bar{\boldsymbol{x}}^t) - \frac{\eta}{2}\mathbb{E}\|\nabla f(\bar{\boldsymbol{x}}^t)\|^2 + \frac{\eta}{2n}\mathbb{E}\|\nabla F(\boldsymbol{X}^t) - \nabla F(\bar{\boldsymbol{x}}^t)\|_{\mathrm{F}}^2 - \left(\frac{\eta}{2} - \frac{\eta^2 L}{2}\right)\mathbb{E}\|\bar{\boldsymbol{v}}^t\|^2 + \frac{\eta\sigma^2}{2bn}$

$\overset{(ii)}{\leq} \mathbb{E}f(\bar{\boldsymbol{x}}^t) - \frac{\eta}{2}\mathbb{E}\|\nabla f(\bar{\boldsymbol{x}}^t)\|^2 + \frac{\eta L^2}{2n}\mathbb{E}\|\boldsymbol{X}^t - \bar{\boldsymbol{x}}^t\mathbf{1}^\top\|_{\mathrm{F}}^2 - \left(\frac{\eta}{2} - \frac{\eta^2 L}{2}\right)\mathbb{E}\|\bar{\boldsymbol{v}}^t\|^2 + \frac{\eta\sigma^2}{2bn},$

where (i) comes from Young's inequality, and (ii) comes from the $L$-smooth assumption (Assumption 2) again. Finally, by substituting definitions of $\Omega_3^t$ and $\Omega_5^t$, we reach

$$\mathbb{E}f(\bar{\boldsymbol{x}}^{t+1}) \leq \mathbb{E}f(\bar{\boldsymbol{x}}^t) - \frac{\eta}{2}\mathbb{E}\|\nabla f(\bar{\boldsymbol{x}}^t)\|^2 + \frac{\eta L^2}{2n}\Omega_3^t - \left(\frac{\eta}{2} - \frac{\eta^2 L}{2}\right)\Omega_5^t + \frac{\eta\sigma^2}{2bn}. \tag{19}$$

**Step 2: constructing the Lyapunov function** By representing

$$\boldsymbol{\Omega}^t = [\Omega_1^t \quad \Omega_2^t \quad \Omega_3^t \quad \Omega_4^t]^\top, \tag{20}$$

Lemma 3 can be written more compactly as

$$\boldsymbol{\Omega}^{t+1} \leq \underbrace{\begin{bmatrix} 1 - \frac{\alpha}{2} + \frac{6\gamma^2 C}{\alpha} & 0 & \frac{6\gamma^2 C}{\alpha} & \frac{6\eta^2}{\alpha} \\ \frac{18\gamma^2 CL^2}{\alpha} & 1 - \frac{\alpha}{2} + \frac{6\gamma^2 C}{\alpha} & \frac{18\gamma^2 CL^2}{\alpha} & \frac{6\gamma^2 C + 18L^2\eta^2}{\alpha} \\ \frac{6\gamma C}{\rho} & 0 & 1 - \frac{\gamma\rho}{2} & \frac{6\eta^2}{\gamma\rho} \\ \frac{18\gamma CL^2}{\rho} & \frac{6\gamma C}{\rho} & \frac{18\gamma CL^2}{\rho} & 1 - \frac{\gamma\rho}{2} + \frac{18L^2\eta^2}{\gamma\rho} \end{bmatrix}}_{=:\boldsymbol{A}} \boldsymbol{\Omega}^t$$

$$+ \underbrace{\begin{bmatrix} \frac{6n\eta^2}{\alpha} \\ \frac{18L^2\eta^2 n}{\alpha} \\ 0 \\ \frac{18n\eta^2 L^2}{\gamma\rho} \end{bmatrix}}_{=:\boldsymbol{b}_1} \Omega_5^t + \underbrace{\begin{bmatrix} 0 \\ \frac{12n}{\alpha} \\ 0 \\ \frac{12n}{\gamma\rho} \end{bmatrix}}_{=:\boldsymbol{b}_2} \frac{\sigma^2}{b}. \tag{21}$$

Define the Lyapunov function

$$\Phi_t = \mathbb{E}f(\bar{\boldsymbol{x}}^t) - f^* + \frac{c_1 L}{n} \cdot \Omega_1^t + \frac{c_2\rho^2}{nL} \cdot \Omega_2^t + \frac{c_3 L}{n} \cdot \Omega_3^t + \frac{c_4\rho^2}{nL}\Omega_4^t$$

$$= \mathbb{E}f(\bar{\boldsymbol{x}}^t) - f^* + \boldsymbol{s}^\top \boldsymbol{\Omega}^t, \tag{22}$$

where

$$\boldsymbol{s} = \begin{bmatrix} \dfrac{c_1 L}{n} & \dfrac{c_2 \rho^2}{nL} & \dfrac{c_3 L}{n} & \dfrac{c_4 \rho^2}{nL} \end{bmatrix}$$

for some constants $c_1, c_2, c_3, c_4$ that will be specified later.

By (21) from Lemma 3 and the descent property (19), we have

$\Phi_{t+1}$
$$= \mathbb{E}f(\bar{\boldsymbol{x}}^{t+1}) - f^* + \boldsymbol{s}^\top \boldsymbol{\Omega}^{t+1}$$
$$\leq \mathbb{E}f(\bar{\boldsymbol{x}}^t) - f^* - \frac{\eta}{2}\mathbb{E}\left\|\nabla f(\bar{\boldsymbol{x}}^t)\right\|^2 + \frac{\eta L^2}{2n}\Omega_3^t - \left(\frac{\eta}{2} - \frac{\eta^2 L}{2}\right)\Omega_5^t + \frac{\eta\sigma^2}{2bn} + \boldsymbol{s}^\top\left(\boldsymbol{A}\boldsymbol{\Omega}^t + \Omega_5^t\boldsymbol{b}_1 + \frac{\sigma^2}{b}\boldsymbol{b}_2\right) \tag{23}$$

$$\leq \Phi_t - \frac{\eta}{2}\mathbb{E}\left\|\nabla f(\bar{\boldsymbol{x}}^t)\right\|^2 - \left(\frac{\eta}{2} - \frac{\eta^2 L}{2}\right)\Omega_5^t + \frac{\eta\sigma^2}{2bn} + (\boldsymbol{s}^\top\boldsymbol{A} - \boldsymbol{s}^\top + \boldsymbol{q}^\top)\boldsymbol{\Omega}^t + \boldsymbol{s}^\top(\Omega_5^t\boldsymbol{b}_1 + \boldsymbol{b}_2\frac{\sigma^2}{b})$$

$$= \Phi_t - \frac{\eta}{2}\mathbb{E}\left\|\nabla f(\bar{\boldsymbol{x}}^t)\right\|^2 + (\boldsymbol{s}^\top\boldsymbol{A} - \boldsymbol{s}^\top + \boldsymbol{q}^\top)\boldsymbol{\Omega}^t - \left(\frac{\eta}{2} - \frac{\eta^2 L}{2} - \boldsymbol{s}^\top\boldsymbol{b}_1\right)\Omega_5^t + \left(\frac{\eta}{2n} + \boldsymbol{s}^\top\boldsymbol{b}_2\right)\frac{\sigma^2}{b},$$

where $\boldsymbol{q} = [0 \quad 0 \quad \frac{\eta L^2}{2n} \quad 0]^\top$. For a moment we assume that there exist some constants $c_1, c_2, c_3, c_4 > 0$ such that

$$\boldsymbol{s}^\top(\boldsymbol{A} - \boldsymbol{I}) + \boldsymbol{q}^\top \leq \boldsymbol{0}, \tag{24a}$$

$$\frac{\eta}{2} - \frac{\eta^2 L}{2} - \boldsymbol{s}^\top\boldsymbol{b}_1 \geq 0, \tag{24b}$$

leading to

$$\Phi_{t+1} \leq \Phi_t - \frac{\eta}{2}\mathbb{E}\left\|\nabla f(\bar{\boldsymbol{x}}^t)\right\|^2 + \left(\frac{\eta}{2n} + \boldsymbol{s}^\top\boldsymbol{b}_2\right)\frac{\sigma^2}{b} \leq \Phi_t - \frac{\eta}{2}\mathbb{E}\left\|\nabla f(\bar{\boldsymbol{x}}^t)\right\|^2 + \frac{36c_4\sigma^2}{c_\gamma Lb\alpha}.$$

The proof is thus completed by recursing the above relation over $t = 0, \ldots, T-1$.

**Step 3: verifying (24)** It boils down to verify (24) is feasible, and it is equivalent to verify there exist parameters $c_1, c_2, c_3, c_4, \gamma, \eta > 0$ satisfying the following matrix inequality:

$$\begin{bmatrix} \boldsymbol{I} - \boldsymbol{A}^\top \\ -\boldsymbol{b}_1 \end{bmatrix} \text{diag}\left[\frac{L}{n}, \frac{\rho^2}{nL}, \frac{L}{n}, \frac{\rho^2}{nL}\right]\begin{bmatrix} c_1 \\ c_2 \\ c_3 \\ c_4 \end{bmatrix} \geq \begin{bmatrix} \boldsymbol{q} \\ \frac{\eta^2 L}{2} - \frac{\eta}{2} \end{bmatrix}.$$

Note that by choosing $\gamma = c_\gamma\rho\alpha$, $\eta = c_\eta\gamma\rho^2/L$, and setting $c_\gamma \leq \frac{1}{6\sqrt{C}}$ and $c_\eta \leq \frac{1}{9}$, we get

$$1 - \frac{\alpha}{2} + \frac{6\gamma^2 C}{\alpha} \leq 1 - \frac{\alpha}{4}, \quad 1 - \frac{\gamma\rho}{2} + \frac{18L^2\eta^2}{\gamma\rho} \leq 1 - \frac{\gamma\rho}{4}. \tag{25}$$

Now, it suffices to show that there exist $c_1, c_2, c_3, c_4, c_\gamma, c_\eta > 0$ such that the following inequalities are satisfied:

$$\begin{bmatrix} \frac{\alpha L}{4n} & -\frac{18c_\gamma^2\alpha\rho^4 L}{n} & -\frac{6c_\gamma\alpha L}{n} & -\frac{18Cc_\gamma\alpha\rho^2 L}{n} \\ 0 & \frac{\alpha\rho^2}{4nL} & 0 & -\frac{6Cc_\gamma\alpha\rho^2}{nL} \\ -\frac{6c_\gamma\rho\gamma CL}{n} & -\frac{18c_\gamma\rho\gamma L}{n} & \frac{\gamma\rho L}{2n} & -\frac{18C\gamma\rho L}{n} \\ -\frac{6c_\eta^2 c_\gamma\gamma\rho^3}{nL} & -\frac{6C\gamma\rho^3(1+3c_\eta^2\rho^4)}{nL} & -\frac{6c_\eta^2\gamma\rho^3}{nL} & \frac{\gamma\rho^3}{4nL} \\ -12c_\eta c_\gamma\rho^3\frac{\eta}{2} & -36c_\eta c_\gamma\rho^5\frac{\eta}{2} & 0 & -36c_\eta\rho^2\frac{\eta}{2} \end{bmatrix}\begin{bmatrix} c_1 \\ c_2 \\ c_3 \\ c_4 \end{bmatrix} \geq \begin{bmatrix} 0 \\ 0 \\ \frac{c_\eta\gamma\rho L}{2n} \\ 0 \\ (c_\eta c_\gamma\alpha\rho^3 - 1)\frac{\eta}{2} \end{bmatrix}.$$

Given $\alpha \leq 1, \rho \leq 1$, this can be further reduced to show the existence of $c_1, c_2, c_3, c_4, c_\gamma, c_\eta > 0$ such that

$$\begin{bmatrix} 1 & -72Cc_\gamma^2 & -24Cc_\gamma & -72Cc_\gamma \\ 0 & 1 & 0 & -24Cc_\gamma \\ -12Cc_\gamma & -35Cc_\gamma & 1 & -36C \\ -24c_\eta^2 c_\gamma & -24c_\gamma(1+3c_\eta^2) & -24c_\eta^2 & 1 \\ -12c_\eta c_\gamma & -36c_\eta c_\gamma & 0 & -36c_\eta \end{bmatrix}\begin{bmatrix} c_1 \\ c_2 \\ c_3 \\ c_4 \end{bmatrix} \geq \begin{bmatrix} 0 \\ 0 \\ c_\eta \\ 0 \\ -1+c_\eta c_\gamma \end{bmatrix}.$$

This can be easily verified by noting that as long as $c_\eta$ and $c_\gamma$ are set sufficiently small, it is straightforward to find feasible $c_1, c_2, c_3, c_4$.

## D.4 Proof of Theorem 3 and 4

Since Theorem 4 is a generalization of Theorem 3, it suffices to directly prove Theorem 4. The proof strategy of Theorem 4 is similar to that of Theorem 2. However, in order to achieve the advertised linear convergence rate under the PL condition (Assumption 4), we need to use a slightly different linear system.

Denote $\kappa := L/\mu$. Taking the same Lyapunov function $\Phi_t$ in (22), by the same argument of Section D.3 up to (23), we have

$$\Phi_{t+1}$$

$$\leq \mathbb{E}f(\bar{\boldsymbol{x}}^t) - f^* - \frac{\eta}{2}\mathbb{E}\left\|\nabla f(\bar{\boldsymbol{x}}^t)\right\|^2 + \frac{\eta L^2}{2n}\Omega_3^t - \left(\frac{\eta}{2} - \frac{\eta^2 L}{2}\right)\Omega_5^t + \frac{\eta\sigma^2}{2bn} + \boldsymbol{s}^\top\left(\boldsymbol{A}\boldsymbol{\Omega}^t + \Omega_5^t\boldsymbol{b}_1 + \frac{\sigma^2}{b}\boldsymbol{b}_2\right)$$

$$\leq (1 - \eta\mu)(\mathbb{E}f(\bar{\boldsymbol{x}}^t) - f^*) + (\boldsymbol{s}^\top\boldsymbol{A} + \boldsymbol{q}^\top)\boldsymbol{\Omega}^t - \left(\frac{\eta}{2} - \frac{\eta^2 L}{2} - \boldsymbol{s}^\top\boldsymbol{b}_1\right)\Omega_5^t + \left(\frac{\eta}{2n} + \boldsymbol{s}^\top\boldsymbol{b}_2\right)\frac{\sigma^2}{b}$$

$$= (1 - \eta\mu)\Phi_t + \left(\boldsymbol{s}^\top\boldsymbol{A} - (1 - \eta\mu)\boldsymbol{s}^\top + \boldsymbol{q}^\top\right)\boldsymbol{\Omega}^t - \left(\frac{\eta}{2} - \frac{\eta^2 L}{2} - \boldsymbol{s}^\top\boldsymbol{b}_1\right)\Omega_5^t + \left(\frac{\eta}{2n} + \boldsymbol{s}^\top\boldsymbol{b}_2\right)\frac{\sigma^2}{b},$$

where $\boldsymbol{q} = [0 \quad 0 \quad \frac{\eta L^2}{2n} \quad 0]^\top$, and the second inequality follows from the PL condition (Assumption 4). If we can establish that there exist there exist some constants $c_1, c_2, c_3, c_4$ such that

$$\boldsymbol{s}^\top(\boldsymbol{A} - (1 - \eta\mu)\boldsymbol{I}) + \boldsymbol{q}^\top \leq \boldsymbol{0}, \tag{26a}$$

$$\frac{\eta}{2} - \frac{\eta^2 L}{2} - \boldsymbol{s}^\top\boldsymbol{b}_1 \geq 0, \tag{26b}$$

we arrive at

$$\Phi_{t+1} \leq (1 - \eta\mu)\Phi_t + \left(\frac{\eta}{2n} + \boldsymbol{s}^\top\boldsymbol{b}_2\right)\frac{\sigma^2}{b} \leq (1 - \eta\mu)\Phi_t + \frac{36c_4\sigma^2}{c_\gamma Lb\alpha}.$$

Recursing the above relation then complete the proof.

It then boils down to establish (26). By similar arguments as Section D.3, in view of (25) and $\alpha \leq 1, \rho \leq 1$, it is sufficient to show there exist constants $c_1, c_2, c_3, c_4, c_\gamma, c_\eta > 0$ such that

$$\begin{bmatrix} 1 - \frac{4c_\eta c_\gamma}{\kappa} & -72Cc_\gamma^2 & -24Cc_\gamma & -72Cc_\gamma \\ 0 & 1 - \frac{4c_\eta c_\gamma}{\kappa} & 0 & -24Cc_\gamma \\ -12Cc_\gamma & -35Cc_\gamma & 1 - \frac{2c_\eta}{\kappa} & -36C \\ -24c_\eta^2 c_\gamma & -24c_\gamma(1 + 3c_\eta^2) & -24c_\eta^2 & 1 - \frac{4c_\eta}{\kappa} \\ -12c_\eta c_\gamma & -36c_\eta c_\gamma & 0 & -36c_\eta \end{bmatrix} \begin{bmatrix} c_1 \\ c_2 \\ c_3 \\ c_4 \end{bmatrix} \geq \begin{bmatrix} 0 \\ 0 \\ c_\eta \\ 0 \\ -1 + c_\eta c_\gamma \end{bmatrix}.$$

This can be easily verified by noting that as long as $c_\eta$ and $c_\gamma$ are set sufficiently small, it is straightforward to find feasible $c_1, c_2, c_3, c_4$.