# OpenReview forum: "BEER: Fast $O(1/T)$ Rate for Decentralized Nonconvex Optimization with Communication Compression"
_NeurIPS.cc/2022/Conference — NeurIPS 2022 Accept_

### Official Review · Reviewer_AhEM · 2022-07-09

**Rating:** 6
**Confidence:** 5
**Soundness:** 3 good
**Presentation:** 3 good
**Contribution:** 3 good

**Summary:**

This paper proposes a decentralized algorithm with communication compression named BEER. In the deterministic non-convex setting, BEER improves the convergence rate of existing algorithms to O(1/T), which is a state-of-the-art result. The author also establish the convergence rate of BEER in the stochastic setting or with PL conditions. Numerical experiments have been conducted to validate the results.

**Questions:**

While the convergence rate established in the deterministic scenario is SOTA, the rate for stochastic settings is not.

1. According to the established rate, BEER cannot converge to the stationary solution even with decaying learning rate. In contrast, other decentralized compression algorithms such as CHOCO-SGD and SQuARM-SGD can converge to the stationary solution with decaying rate. It is a major drawback;

2. The authors argue that by utilizing large enough mini-batches, BEER can converge to the stationary solution. In this setting, its convergence rate is still worse than CHOCO-SGD and SQuARM-SGD if they also utilize large enough mini-batches.

3. Another major drawback is that, under the stochastic setting, BEER does not show a linear speedup rate. In contrast, SQuARM-SGD shows a clear linear speedup term $O(\sigma^2/\sqrt{nT})$. In contrast, BEER cannot even have a diminishing term associated with $\sigma^2$.

4. Also, even in the deterministic non-convex setting, the dependence on the network topology established in existing works (without communication compression) is far better than $O(1/\rho^3)$. For example, in the following reference, the network topology dependence of the GT algorithm is $O(1/\rho)$.

[Ref] A.  Koloskova, T. Lin, and S. Stich, An Improved Analysis of Gradient Tracking for Decentralized Machine Learning, NeruIPS 2021.

Can the authors please have a few comments on the above points. Are these drawbacks valid? Can these drawbacks be improved? If not, can the authors provide the reason or insight why these drawbacks are difficult to overcome?


**Limitations:**

Yes

**Strengths And Weaknesses:**

### Strength

- BEER works under weaker assumptions. It removes the assumption on bounded gradients and gradient dissimilarity;


- In the deterministic setting, BEER achieves the state-of-the art convergence rate in terms of T.

### Weakness

- The convergence rate in the stochastic and non-convex setting is not satisfactory; it is worse than existing algorithms such as SQuARM-SGD

- The dependence on the network topology is not satisfactory; it is worse than existing state-of-the-art algorithms

In general, the reviewer values the strengths more and hence rates the paper positively.

---

> ### Author Response · Authors · 2022-08-02
> **Response to Reviewer AhEM**
>
> Dear Reviewer AhEM,
>
> Thanks for the positive evaluation of our paper as well as the insightful suggestions and feedback! In particular, we appreciate acknowledging that BEER achieves the state-of-the-art rate $O(1/T)$ for the deterministic setting without the bounded gradients and gradient dissimilarity. Below is a point-to-point response to your questions.
>
> ## General response about the rates in the stochastic case
> We want to begin by highlighting that the rate of BEER is the only existing bound that does not suffer from unbounded gradients and gradient dissimilarity, where existing bounds might become vacuous when $G$ goes to infinity (please see the following table for more details).
>
> | Algorithm | Convergence rate | Gradient complexity when $\sigma^2>0$ | Assumptions|
> | --- | --- | --- | --- |
> | SQuARM-SGD | $O(1/\sqrt{nT} + nG^2 / T)$ | $O(\sigma^2 / n\epsilon^4 + nG^2 / \epsilon^2)$ | Bounded gradient |
> | Deepsqueeze | $O((G/T)^{2/3})$ | $O(\sigma^2 / n\epsilon^4 + G / \epsilon^3)$ | Bounded dissimilarity |
> | CHOCO-SGD | $O((G/T)^{2/3})$ | $O(\sigma^2 / n\epsilon^4 + G / \epsilon^3)$ | Bounded gradient |
> | BEER | $O(1/T)$ | $O(\sigma^2 / \epsilon^4 + 1 / \epsilon^2)$ | --- |
>
> With this unique feature, our rate of BEER has different characteristics and potential weaknesses in other aspects (some are also shared by existing works). We believe some of these drawbacks might be improvable, but some might be challenging from a technical perspective as we shall detail further below.

---

> > ### Comment · Reviewer_AhEM · 2022-08-05
> > **Your response is very confusing**
> >
> > \
> > First, I would thank the authors for their detailed replies. However, some arguments in the reply are very confusing. They do not clarify my concerns.
> >
> > \
> > Non-vanishing term $\sigma^2$ is an obvious weakness in BEER.  CHOCO-SGD and SQuARM-SGD do not suffer from these issues. The reviewer mentioned the LEAD paper (Linear Convergent Decentralized Optimization with Compression, Liu et al, ICLR 2021). However, I don't see any non-vanishing terms in the rate. For example, in Eq. (11) in https://arxiv.org/pdf/2007.00232.pdf, the variance-related term is $\eta^2 \sigma^2$, rather than $\eta \sigma^2$ listed in your response. When decaying $\eta$ is used, LEAD can converge to the exact solution. The existence of LEAD brings up a severe concern: is BEER a superior method to LEAD? LEAD does not have data-heterogeneity assumption, and it converge better than BEER in PL (strongly-convex) conditions without any non-vanishing terms. I now come to question the contribution of BEER. Please compare carefully with LEAD. It seems BEER only considers an additional non-convex case, which by itself cannot make it accepted into NeurIPS.
> >
> > \
> > The arguments on linear speedup is also confusing. In my opinion, if the leading term in the convergence rate involves $nT$, we say this algorithm can achieve (asymptotic) linear speedup. When $T$ is sufficiently large (which is very practical since deep learning typically takes many iterations), both CHOCO-SGD and SQuARM-SGD can achieve linear speedup. In contrast, the proposed BEER can never achieve linear speedup even for sufficiently large $T$.
> >
> > \
> > As to the network topology dependence, it holds easily that $O(1/(c\rho^2)) = O(1/\rho^2)$. In the paragraph below Remark 1 in (https://arxiv.org/pdf/2202.03836.pdf), it says for any given $W$, if we use $\widetilde{W} = (I+W)/2$ in the algorithm, $c=1$ is ensured easily. However, even if $\widetilde{W} = (I+W)/2$ is used in BEER, the network topology dependence cannot be improved.
> >
> > \
> > I am considering lowering the rating now. Hope the authors can clarify the above concerns (especially the first one) clearly.

---

> > > ### Author Response · Authors · 2022-08-06
> > > **Further clarifications of our results (1/2)**
> > >
> > > Dear reviewer,
> > >
> > > Thanks for your feedback.
> > >
> > > First, we would like to point out that there are two complexities (communicaiton complexity and gradient complexity). It seems that there are some mixups of these two complexities from your comment which thus lead to some misunderstanding regarding the comparisons between BEER and other works.
> > > Now to make it clear, let us explain in more details, first by setting up the complexity measures to make sure we are on the same page.
> > >
> > > > 1. Communiation complexity:= communication rounds $\times$ communication cost per round.
> > >
> > > Note that we only need to focus on *communication rounds* since the communication cost per round are the same for all compressed algorithms under the same compression operator.
> > >
> > > > 2. Gradient complexity: = Communication rounds $\times$ number of gradient computations per round.
> > >
> > > Now we emphsize that our main contribution is to improve the *communication complexity* for this decentralized nonconvex optimization.
> > >
> > > For achieving $\epsilon$-accuracy solution, the communicaiton rounds for algorithms are as follows:
> > >
> > >
> > > | Algorithms | Communication rounds | Assumptions |
> > > | --- | --- | --- |
> > > | SQuARM-SGD | $O(\frac{nG}{\epsilon^2} + \frac{\sigma^2}{bn\epsilon^4})$ | Bounded gradient |
> > > | DeepSqueeze| $O(\frac{G}{\epsilon^3} + \frac{\sigma^2}{bn\epsilon^4})$ | Bounded dissimilarity |
> > > | CHOCO-SGD  | $O(\frac{G}{\epsilon^3} + \frac{\sigma^2}{bn\epsilon^4})$ | Bounded gradient |
> > > | BEER | $O(\frac{1}{\epsilon^2})$ for deterministic or large minibatch| --- |
> > >
> > > * $n$ is the number of agents, $\epsilon$ is the convergence error, $\sigma^2$ is the variance of local stochatic gradient $\mathbb{E}[ \|\tilde{\nabla}_b f_i(x) - \nabla f_i(x)\|^2] \leq \frac{\sigma^2}{b}$, where ${\tilde{\nabla}}_b f_i(x)= \frac{1}{b}\sum_\{j=1\}^b \nabla f_i(x;\xi_\{i,j\})$ is the minibatch (with size $b$) stochastic gradient on agent $i$.
> > > * $G$ denotes the bounded gradient/dissimilarity assumption ($\mathbb{E}_\{\xi_\{i,j\}\}\|\nabla f_i (x;\xi_\{i,j\})\|^2 \leq G^2$ or $\mathbb{E}_i\|\nabla f_i(x) - \nabla f(x)\|^2 \leq G^2$). It is not hard to see that $\sigma^2\leq G^2$ for bounded gradient $G$.
> > > * Our BEER result $O(\frac{1}{\epsilon^2})$ holds for deterministic or large minibatch cases, where deterministic means that agent computes its local full gradient (then $\sigma^2=0$) if there is few data samples on the agent, and large minibatch means that agent computes $b=O(\frac{\sigma^2}{\epsilon^2})$ minibatch stochastic gradients otherwise. Note that in both cases, previous  algorithms still require at least $O(\frac{G}{\epsilon^3})$ communication rounds, which is much worse than $O(\frac{1}{\epsilon^2})$, i.e., $(\frac{G}{T})^{\frac{2}{3}}$ vs. $\frac{1}{T}$.
> > >
> > > >In sum, our BEER achieves much better communication complexity than previous work. None of them can achieve $O(\frac{1}{T})$ (i.e. $O(\frac{1}{\epsilon^2})$ communciation rounds) even if $\sigma^2=0$ or agents use any minibatch size $b$.
> > >
> > > Now, let's look at the gradient complexity.
> > >
> > > | Algorithms | Communication rounds | Number of gradient computations per round | Assumptions |
> > > | --- | --- | --- | --- |
> > > | SQuARM-SGD | $O(\frac{nG}{\epsilon^2} + \frac{\sigma^2}{bn\epsilon^4})$ | $b\geq 1$ (minibatch size) |  Bounded gradient |
> > > | DeepSqueeze| $O(\frac{G}{\epsilon^3} + \frac{\sigma^2}{bn\epsilon^4})$ | $b \geq 1$ |  Bounded dissimilarity |
> > > | CHOCO-SGD  | $O(\frac{G}{\epsilon^3} + \frac{\sigma^2}{bn\epsilon^4})$ | $b \geq 1$ | Bounded gradient |
> > > | BEER | $O(\frac{1}{\epsilon^2})$ for deterministic or large minibatch| $b = O(\frac{\sigma^2}{\epsilon^2})$ | --- |
> > >
> > > * Previous algorithm can use small minibach size $b$, however the choice of $b$ will not change/improve the first term ($\frac{nG}{\epsilon^2}$ or $\frac{G}{\epsilon^3}$) in their communication rounds. So it is not hard to see that $b=1$ achieves their best gradient complexity (i.e. communication rounds $\times$ number of gradient computations per round).
> > > * As we mentioned above, our BEER requires a large minibatch $b=O(\frac{\sigma^2}{\epsilon^2})$, so our gradient complexity $O(\frac{1}{\epsilon^2} \cdot (\frac{\sigma^2}{\epsilon^2} +1))$ may be worse than previous work such as $O(\frac{G}{\epsilon^3} + \frac{\sigma^2}{n\epsilon^4})$ of CHOCO-SGD.
> > >
> > >
> > > > In sum, our gradient complexity $O(\frac{\sigma^2}{\epsilon^4} + \frac{1}{\epsilon^2})$ might be worse than previous result $O(\frac{G}{\epsilon^3} + \frac{\sigma^2}{n\epsilon^4})$. Note that the linear speedup $n$ only improves the second term $\frac{\sigma^2}{n\epsilon^4}$ and the gradient complexity is still at least $O(\frac{G}{\epsilon^3})$.

---

> > > > ### Comment · Reviewer_AhEM · 2022-08-08
> > > > **Reply**
> > > >
> > > > \
> > > > Many thanks for the careful response.The benefits and drawbacks of the proposed BEER algorithm get much clearer now.
> > > >
> > > > \
> > > > The authors may consider involving the above rebuttal discussions into their manuscript in later revisions. In particular,
> > > >
> > > > 1. The comparison with existing works in terms of communication complexity and gradient complexity
> > > >
> > > > 2. The comparison with LEAD in the strongly convex (since strongly convex is a special case of PL) case
> > > >
> > > > 3. Remove the arguments in lines 202 - 205 because gradient tracking, according to the recent studies, does not have a worse dependence on network topology than DSGD
> > > >
> > > > 4. Discussions on linear speedup

---

> > > > > ### Author Response · Authors · 2022-08-09
> > > > > **Re: Reply**
> > > > >
> > > > > Dear Reviewer AhEM,
> > > > >
> > > > > We are happy to hear that our clarifications of results are clear to you now.
> > > > > If you think our responses have addressed your concerns well, we will appreciate if you could keep and consider increasing your support, and we will be happy to answer any other questions if you have.
> > > > >
> > > > > Thanks for the further revision suggestions. We will add these comparisons and discussions, and remove the arguments in the updated version.
> > > > >
> > > > > Authors

---

> > > ### Author Response · Authors · 2022-08-06
> > > **Further clarifications of our results (2/2)**
> > >
> > > Now, let's make a final remark on our contributions.
> > > > Our BEER **always achieves much better communication complexity** than previous work. None of them can achieve $O(\frac{1}{T})$ (i.e. $O(\frac{1}{\epsilon^2})$ communciation rounds). Note that in the decentralize/multi-agent setting where the communication is one of the main bottlenecks, thus BEER makes a significant improvement. Moreover, we also **remove the strong bounded gradient/dissimilarity assumption**, thus BEER allows arbitrary data heterogeneity which is clearly an important feature for multi-agent learning.
> > >
> > > > For the drawback of our BEER, it may lead to a worse gradient complexity which is due to the large minibatch size (also note that this only affects local computation cost, and does not affect communication rounds at all). However, we would like to mention that the large minibatch size $b$ is just for theoretical analysis and it is possible that people use small $b$ as other algorithms in practice, while still maintaining fewer communication rounds (this is also observed in our numberical experiments).
> > >
> > > Finally, let's get back and answer your concerns one by one in more detail.
> > >
> > > Regarding the non-vanishing term $\sigma^2$, as we discussed above and also in Theorem 2, it is clearly that this $\sigma^2$ term will disappear (or reduce to ignorable $\epsilon$-accuracy level) for deterministic case (or large minibatch case).
> > > The result for PL condition is just a byproduct, not the main result of BEER. Also note that PL condition is still in the **nonconvex** setting and thus is not fair to compare it with strongly convex setting/result (Strong convexity implies PL condition but not the other way around; so existing results on the strongly convex case might not hold for the PL case).
> > >
> > >
> > > Regarding the linear speedup, it only improves the second term of the gradient complexity as we mentioned above. Based on discussions above, we do not think this point will hurt our main contributions (communicaiton complexity).
> > >
> > >
> > > In our original response, we said that *"We believe that by applying their fine-grained analysis, it is possible to improve the convergence rate of BEER from $O(1/\rho^3)$ to $O(1/(c\rho^2))$."* We did not claim that our BEER can enjoy the better dependency on $\rho$ right now. Also the transform of $W$ to $\widetilde{W}$ as you mentioned will change the algorithm and the analysis, so it is not clear for BEER whether the improvment will work or not.
> > >
> > >
> > > >We believe our response above has clearly explained our main contributions (enjoy much fewer communication rounds & remove strong bounded gradient/dissimilarity assumption), and should have addressed your concerns.
> > > We do not think these minor points (only affect local computation cost) weaken our main contributions (better communcation complexity and allowing arbitrary data heterogeneity among agents) for this decentralized nonconvex setting.
> > >
> > >
> > > Respectfully,
> > > Authors

---

> ### Author Response · Authors · 2022-08-02
> **Continues to respond Reviewer AhEM**
>
> ## Non-diminishing terms on $\sigma^2$
> Our stochastic bound indeed includes non-diminishing terms on $\sigma^2$, which we currently do not have a very good idea to deal with. While the bound suggests a large minibatch is needed, we do not observe the need for a large minibatch in the numerical experiments, and therefore it is plausible that this term is an artifact of the proof techniques. In particular, we believe that these non-diminishing terms may come from the combination of compression and gradient tracking (GT); for example, the LEAD algorithm (Linear Convergent Decentralized Optimization with Compression, Liu et al, ICLR 2021), which also combines compression and GT-like ideas, include this non-diminishing variance term as well. Specifically in our proof, this term directly comes from the marriage of GT and the biased compressed operator. Note that to get the diminishing term on $\sigma^2$, one needs to obtain $\eta^2 \sigma^2$ in the recursion of the Lyapunov function, where $\eta$ is the learning rate. However because we use the biased compressor, we can only get $\eta \sigma^2$ in the Lyapunov function recursion. Specifically, we only get something like
> $$ \Phi^{(t+1)} \le \Phi^{(t)} - \eta \| \nabla f(x^{(t)}) \|^2 + \eta \sigma^2, $$
> where we omit constants and other dependencies. Thus, this term is non-diminishing in our proof, and we need to use a large batch size to deal with this term. It is an interesting future direction to improve this dependency.
>
> ## Performance under large mini-batch
> We need to clarify that when all algorithms use the large mini-batch, BEER has a faster convergence rate compared with CHOCO-SGD and SQuARM-SGD. Even though all algorithms use the full gradient computation, CHOCO-SGD and SQuARM-SGD still have the $O((G/T)^{2/3})$ convergence rate while BEER only has $O(1/T)$. However, the gradient complexity of BEER may be worse than that of CHOCO-SGD and SQuARM-SGD (in the very heterogeneous setting where $G$ is very large, BEER has an even smaller gradient complexity).
>
> Besides, we also want to emphasize that the convergence rate affects the communication cost, while the gradient complexity only affects the local computations. In the federated learning/decentralized setting where communication is the main bottleneck, our BEER can converge with much less communication.
>
> ## Discussions on linear speedup
> First, we want to clarify that even CHOCO-SGD and SQuARM-SGD do not achieve linear speedup on $\sigma^2$ in terms of the number of clients in general. In Theorem 2 of the SQuARM-SGD paper (https://arxiv.org/pdf/2005.07041.pdf), the convergence rate in the nonconvex setting includes terms like $$\sigma^2 / \sqrt{nT} + nG^2 / T$$, where we omit other terms and dependencies on the spectral gap and compression parameter. Here, $G$ is the bounded gradient parameter and note that it is greater than $\sigma$ (equality holds when all clients are homogeneous). Thus, it does not have any speed up in terms of the number of clients, when $\sigma^2 /\sqrt{nT}$ is not the dominating term, which happens either when the number of clients is large or the number of iterations is not too large. Our intuition is that all these methods, e.g. CHOCO-SGD, SQuARM-SGD, and our algorithm use the \textbf{biased} compression operators, and in general \textbf{biased} compression operators are hard to enjoy this linear speed up (it can only speed up in certain cases).
>
> ## Dependency on the network topology
> Thanks for bringing up the paper [Koloskova et al, NeurIPS 2021]! In their analysis, they managed to improve a factor of the dependency with $\rho$, by replacing it with another network topology parameter $c$, which in the worst case goes to $\rho$. Thus, [Koloskova et al, NeurIPS 2021] can be regarded as a fine-grained analysis of the original GT proof, whose worst-case convergence rate is still $1/(c\rho) = 1/\rho^2$ for GT algorithms (see Theorem 2 and Section 5.1 in Koloskova et al, NeurIPS 2021). We believe that by applying their fine-grained analysis, it is possible to improve the convergence rate of BEER from $O(1/\rho^3)$ to $O(1/(c\rho^2))$. However, further improvements appear to be hard because all these existing algorithms with compression seem to have a worse dependency on $\rho$ compared to the original uncompressed versions (Line 201-206 on page 6).

---

### Official Review · Reviewer_3QWf · 2022-07-09

**Rating:** 6
**Confidence:** 4
**Soundness:** 2 fair
**Presentation:** 3 good
**Contribution:** 3 good

**Summary:**

This paper presents a new decentralized nonconvex optimization algorithm, termed BEER, with communication compression to reduce the number of communication rounds in networked systems. Specifically, the authors introduced two auxiliary variables to track the parameter and global gradient updates. They theoretically showed that the convergence rate was improved over the state-of-the-art and matched the uncompressed counterpart. Additionally, they also analyzed the convergence when the objective satisfies the PL condition. They claimed that the strong bounded gradient assumption can be relaxed using the proposed algorithm. To validate BEER, the authors used two simple benchmark datasets and two simple models to show that the proposed algorithm is comparable to the uncompressed algorithm and better than the best available compressed decentralized algorithm.

**Questions:**

1. Is $\Phi_T$ with respect to $T$? Can the authors show it is a constant in the error bound in both Theorem 1 and Theorem 2?
2. How is the performance of the proposed algorithm when it is applied to larger benchmark datasets and more complex models?

**Limitations:**

I don't think the authors have adequately addressed the limitations of their proposed approach. It would be better and great to see some discussion regarding this aspect, particularly in terms of the perspective of applications and the comparison with other existing methods.

**Strengths And Weaknesses:**

I think overall the investigated topic in this paper is quite interesting, as communication overhead is a critical issue that needs to be resolved in order to achieve scalability in multi-agent systems. This paper is well written and easy to follow. While I still have a few concerns here that the authors should address to make the current paper more technically solid and sound.

1. The algorithmic framework has to be agent-wise. Though the authors have defined the terminologies well and explain the algorithm clearly. Since the focus of this paper is a decentralized optimization algorithm, it will be clear and straightforward to understand how the parameters are updated. That way, an agent-wise algorithm is better than the current algorithmic framework.

2. The authors defined the Lyapunov function, $\Phi_t$ , to measure the convergence progress. For Theorem 1 and Theorem 2, the error bounds are actually determined by $\Phi_T$, which might be with respect to $T$. Hence, it will cancel out the $T$ in the denominator. I wonder if the authors could get something like $\Phi_0-\Phi_T\leq \Phi_0-\Phi^{*}$, which will turn the error bound being independent of $\Phi_T$. This is typically in theoretical analysis. Otherwise, the authors need to make a guarantee that $\Phi_T$ won’t be with respect to $T$ in the paper. Additionally, by inspecting Theorem 3 and Theorem 4, the authors use $\Phi_t$ as the measure. Since you have used $f$ in the previous two theorems, why didn’t you follow the same idea?

3. The experimental results are not promising. Though the theoretical analysis looks interesting and decent, to me, the experimental results are not that convincing, due to the only usage of very simple datasets and models. Given the results by baseline methods, BEER actually performs comparably. I think the authors should be presenting more quantitatively to tell how much of gain BEER can achieve. The current discussion is too qualitative. It should be noted that these experiments were conducted by only using simple datasets and models. Then how about with more complicated datasets, such as CIFAR 10, CIFAR 100, ImageNet, and models, such as VGG and ResNet? Moreover, the authors have mentioned different other decentralized optimization algorithms with communication compression in the paper? Then why not compare BEER with them? The authors should present more experimental results to validate the proposed algorithm.

4. Regarding the coefficients for $c_1, c_2, c_3, c_4$ in the Lyapunov function, the authors mentioned in the paper that they depend on different theorems, can the authors give specific values at least for one scenario?

5. Minor comment: The authors have used $\alpha$ as the compression constant. Then for the nonconvex regularization, they should have used another different character.

***********************************************
Overall, the responses from the authors clarified my confusion and addressed most of my concerns. While I think this work delivers some useful insights to the community in theory, which is not a reason to just have simple evaluations for validating the proposed method. The authors have claimed that they will have additional experimental results, which can add values to the paper. So I raised my score. However, the authors should take all comments from other reviewers into account to make the draft more technically solid and sound.

---

> ### Author Response · Authors · 2022-08-02
> **Response to Reviewer 3QWf**
>
> Dear Reviewer 3QWf,
>
> Thanks for the positive evaluation of our paper as well as the insightful suggestions and comments! Below is a point-to-point response to your questions.
>
> ## An agent-wise algorithmic framework
> Thanks for this nice suggestion! Our current Algorithm 1 is presented in matrix-vector notation, which compactly incorporates the updates of all agents inline with the analysis. We have described the operations of each agent in the main text in Section 3. In the revision, we plan to include another algorithm box that describes the agent-wise operation to facilitate understanding, which will correspond to simply the column-wise update of all parameters in Lines 3-8 of Algorithm 1.
>
> ## Questions about the Lyapunov function
> Note that the Lyapunov function $\Phi_t$ (defined in (4)) is nonnegative for at all iterates $t$, therefore, it follows that the term in Theorem 1 and Theorem 2 satisfies $\frac{\Phi_0 -\Phi_T}{T} \leq \frac{\Phi_0}{T}$, which decays at the rate $1/T$, since $\Phi_0$ depends on the initialization and is independent to $T$. Turning to Theorem 3 and Theorem 4, we are interested in establishing the linear convergence of BEER under the PL condition (Assumption 4). Since the linear convergence is established on the function value gap for the PL case, rather than the gradient norm as in Theorem 1 and 2 for the general nonconvex case, it is easier to directly obtain the linear convergence of the Lyapunov function which contains the function value gap (see (4)). Note that the convergence on the function value gap is stronger than the convergence of the gradient norm, since it implies global convergence which is the benefit when the PL condition is satisfied.
>
> ## Numerical experiments on training large neural network models
> It is worth emphasizing that our work focuses on developing theoretical convergence guarantees from an optimization theory perspective, which often does not align empirically with explaining the performance in training large neural networks. We agree it will be interesting to train deep neural networks \emph{in data heterogeneous setting} with communication compression, however, typical tricks (such as batch normalization) may fail or deteriorate under the data-heterogeneous setting, which is the focus of our setting. Because of the competing considerations of optimization and generalization in training deep neural networks, we find such additional experiments might be beyond the scope of our work.
>
>
> ## Coefficients in the Lyapunov function
> We note that the specific values of these coefficients are established via the feasibility of a set of linear matrix inequalities, see Line 672 and Line 674 at the end of Appendix D.4. While it is straightforward to verify their existence, it is not very straightforward, nor informative, to write out these constants: they tend to be excessively large or small for the ease of proof, and do not closely track the actual performance in practice. For this reason, we prefer to not specify these constants, which are standard in such theoretical analysis.
>
> ## Double use of the notation $\alpha$
> Thanks for catching it! We will use a different notation for the nonconvex regularization in the revision.

---

> > ### Comment · Reviewer_3QWf · 2022-08-07
> > **Feedback #1**
> >
> > Thanks for the responses and clarification from the authors. While I think the answers have addressed most of my concerns, in my opinion, additional experimental results are required in order to make the current draft more technically sound and solid.

---

> > > ### Author Response · Authors · 2022-08-07
> > > **Re: Feedback #1**
> > >
> > > Dear Reviewer 3QWf,
> > >
> > > Thanks for acknowledging that "I think the answers have addressed most of my concerns", much appreciated!
> > >
> > > We are happy to perform additional experiments; this is not a problem - what experiments do you have in mind, and why? Please note that this is a theoretical paper, whose main import are the new and strong theoretical results. As such, experiments are mainly designed to supplement the key results, and any weaknesses due to the volume of experiments, we believe, should be seen as minor. Empirical works require a large number of experiments in order to to provide more evidence that the methods work in various settings. We offer theory which proves that our methods will work. So, fewer experiments should suffice.
> > >
> > > If this is the only remaining concern, would you agree it is minor? Moreover, we are happy to add new experiments.
> > >
> > > Thanks for your consideration!
> > >
> > > Authors

---

### Official Review · Reviewer_bXnv · 2022-07-12

**Rating:** 6
**Confidence:** 4
**Soundness:** 3 good
**Presentation:** 3 good
**Contribution:** 3 good

**Summary:**

In this paper, the authors propose a decentralized nonconvex optimization algorithm with communication compression. With the gradient tracking technique, the convergence rate is improved over the existing ones.

**Questions:**

1. There is another very relevant work that also considers decentralized optimization with communication compression:
Y. Liao, Z. Li, K. Huang, and S. Pu. Compressed gradient tracking methods for decentralized optimization with linear convergence. arXiv preprint arXiv:2103.13748, 2021.
Please clarify the differences (for example, strongly convex and nonconvex).
2. The comparison in Table 1 is unfair. The other algorithms are stochastic but BEER is deterministic.


**Ethics Review Area:**

["I don’t know"]

**Limitations:**

N/A.

**Strengths And Weaknesses:**

Strengths:
1. In this paper, the authors propose a decentralized nonconvex optimization algorithm with communication compression.
2. With the gradient tracking technique, the convergence rate is improved over the existing ones.
3. Overall, the writing is smooth, although there are some typos.

Weaknesses:
1. There is another very relevant work that also considers decentralized optimization with communication compression:
Y. Liao, Z. Li, K. Huang, and S. Pu. Compressed gradient tracking methods for decentralized optimization with linear convergence. arXiv preprint arXiv:2103.13748, 2021.
Please clarify the differences (for example, strongly convex and nonconvex).
2. The comparison in Table 1 is unfair. The other algorithms are stochastic but BEER is deterministic.
3. The numerical experiments could be strengthened if the authors consider training a deeper neural network, and show the evolution of accuracy with respect to runtime.

---

> ### Author Response · Authors · 2022-08-02
> **Response to Reviewer bXnv**
>
> Dear Reviewer bXnv,
>
> Thanks for the positive evaluation of our paper as well as the insightful suggestions and comments! Below is a point-to-point response to your questions.
>
> ## Difference between (Liao et al) and our work
> Indeed, (Liao et al) is quite relevant to ours which also considers decentralized optimization with communication compression and gradient tracking techniques. However, there are several key differences. First, as noted by you, the settings are very different: we consider the nonconvex setting whereas (Liao et al) only considers the strongly convex setting, which can be subsumed by our result under the PL condition. Second, our results also cover the stochastic setting, which supports local clients to use a minibatch instead of the full gradient. Third, the proof in both papers are also quite different, and our nonconvex analysis is not a simple extension of the strongly convex case. Besides, our proof allows us to explicitly compute the dependency on $\rho$ clearly and simply, where the analysis in (Liao et al) does not delineate a clear dependency on $\rho$.
>
> ## Comparison in Table 1
> From Theorem 2 (Line 210), our BEER algorithm also supports local variance and therefore the stochastic setting like other algorithms. In the original Table 1, to highlight the main contribution of BEER, which is to achieve the $O(1/T)$ rate even when the bounded dissimilarity assumption (the $G$ parameter in CHOCO-SGD and other algorithms) is dropped, we have chosen to drop the local variance term to simplify the presentation. We agree with your suggestion to include both the deterministic and stochastic rates for a more complete comparison, and we have included the following updated table for your reference. It is again worth noticing that our bound in the stochastic setting is the only one that does not suffer from an unbounded $G$. Note that $G$ is at least $\sigma$ for CHOCO-SGD and SQuARM-SGD since it is the bounded gradient parameter, and $G$ can be very large in the heterogeneous setting.
>
>
> | Algorithm | Convergence rate | Gradient complexity when $\sigma^2>0$ | Assumptions|
> | --- | --- | --- | --- |
> | SQuARM-SGD | $O(1/\sqrt{nT} + nG^2 / T)$ | $O(\sigma^2 / n\epsilon^4 + nG^2 / \epsilon^2)$ | Bounded gradient |
> | Deepsqueeze | $O((G/T)^{2/3})$ | $O(\sigma^2 / n\epsilon^4 + G / \epsilon^3)$ | Bounded dissimilarity |
> | CHOCO-SGD | $O((G/T)^{2/3})$ | $O(\sigma^2 / n\epsilon^4 + G / \epsilon^3)$ | Bounded gradient |
> | BEER | $O(1/T)$ | $O(\sigma^2 / \epsilon^4 + 1 / \epsilon^2)$ | --- |
>
> ## Numerical experiments on deep neural network
> It is worth emphasizing that our work focuses on developing theoretical convergence guarantees from an optimization theory perspective, which often does not align empirically with explaining the performance in training large neural networks. We agree it will be interesting to train deep neural networks in data heterogeneous setting with communication compression, however, typical tricks (such as batch normalization) may fail or deteriorate under the data-heterogeneous setting, which is the focus of our setting. Because of the competing considerations of optimization and generalization in training deep neural networks, we find such additional experiments might be beyond the scope of our work.

---

> > ### Comment · Reviewer_bXnv · 2022-08-04
> > **Update**
> >
> > I thank the authors for the replies, and would like to keep the score of 6.

---

> > > ### Author Response · Authors · 2022-08-04
> > > **Re: Update**
> > >
> > > Thanks for enaging with us!
> > >
> > > Did we adress all weaknesses and answer all questions satisfactorily? If not, which ones and why? Happy to explain more.
> > >
> > > authors

---

### Meta-Review · Area_Chair_SUY7 · 2022-08-30

**Recommendation:** Accept
**Confidence:** Certain

**Metareview:**

The paper has been considered by two ACs--the following is the (adapted) meta-review after the discussions between the two ACs and the senior AC.

The paper focuses on designing communication-compressed algorithms for decentralized nonconvex optimization, where the clients are only allowed to communicate a small amount of quantized information with their neighbors over a predefined graph topology. With quantization, the best known achievable convergence rate for this scenario (i.e. consensus converge to a ) was O(T^{-2/3}). This paper improves this rate to O(T^(-1)) via a novel algorithm called BEER. The main idea behind BEER is to track the model estimate as well as the overall gradient through the message-passing rounds that helps to reduce the effect of the errors due to compression. It is noteworthy that BEEr does not need any extra assumption on the objective (e.g. bounded gradients, etc). Also, BEER is matching the rate without compression even under arbitrary data heterogeneity. Hence, the paper provides a set of interesting and notable results in the field of decentralized learning.

Overall, all the reviewers were in favor of (weakly) accepting the paper. Most of the concerns that the authors had were alleviated in the discussion phase. The main weakness that was discussed among the reviewers was the lack of thorough experimental results. However, we believe that the paper should be judged by its theoretical merits. Even though the paper's main contribution is theoretical, I strongly recommend to the authors to show thew superiority of their proposed method on practical models/settings/data compared to SOTA methods such as Seep Squeeze, CHOCO-SGD, or SQuARM-SGD (please see the reviews for more details). Such experiments would showcase the applicability of BEER in practical settings.

Finally, I would also like to point out that the idea of gradient tracking has been used in Federated learning to improve the rates (in the presence of quantized messages). E.g. see FedLin by Mitra et al (2021), or SCAFFOLD by Karimireddy et all (2020). These papers show for example that gradient tracking can help to get similar rates when the functions are strongly convex (which is analogous to the Polyak condition considered in this submission). I recommend that the authors add a discussion in the final version about the relevant to these works.


**Award:**

No

---

### Decision · Program_Chairs · 2022-09-14

Accept